# Flexible specificity of memory in *Drosophila* depends on a comparison between choices

**Mehrab N Modi[1], Adithya E Rajagopalan[1,2], Hervé Rouault[1,3], Yoshinori Aso[1], Glenn C Turner[1]***

[1]HHMI Janelia Research Campus, Ashburn, United States; [2]The Solomon H. Snyder Department of Neuroscience, Johns Hopkins University School of Medicine, Baltimore, United States; [3]Aix-Marseille Univ, Université de Toulon, CNRS, CPT (UMR 7332), Turing Centre for Living Systems, Marseille, France

**Abstract:** Memory guides behavior across widely varying environments and must therefore be both sufficiently specific and general. A memory too specific will be useless in even a slightly different environment, while an overly general memory may lead to suboptimal choices. Animals successfully learn to both distinguish between very similar stimuli and generalize across cues. Rather than forming memories that strike a balance between specificity and generality, *Drosophila* can flexibly categorize a given stimulus into different groups depending on the options available. We asked how this flexibility manifests itself in the well-characterized learning and memory pathways of the fruit fly. We show that flexible categorization in neuronal activity as well as behavior depends on the order and identity of the perceived stimuli. Our results identify the neural correlates of flexible stimulus-categorization in the fruit fly.

## Editor's evaluation

Memory recall is more precise when discrimination is required. This work in *Drosophila* shows that two related odors trigger near identical Kenyon cell responses when tested in isolation, but trigger different responses to the second odor if these are experienced in sequence within a small temporal window. The authors argue that this template comparison requires some activity downstream of Kenyon cells, that is recruited by MBONs. Overall, the experiments, building on a clever method to build "miminal memories" via optogenetically restricting the formation of memory traces in selective output compartments of the Kenyon cell (KC) axon terminals, provide very nice physiological evidence for a neural mechanism that underlies a contextual basis for the precision of memory recall.

**\*For correspondence:**
turnerg@janelia.hhmi.org

**Competing interest:** The authors declare that no competing interests exist.

## Introduction

Animals routinely encounter competing options and must select between them to survive in complex environments. Making such a choice requires assigning options with subjective values that can be updated through learned experience (*Hare et al., 2011*; *Hunt et al., 2012*; *Glimcher and Fehr, 2013*). In such a framework, storing and updating values for every potential option separately would be computationally taxing (*Seger and Miller, 2010*). Instead, it is beneficial for the brain to maintain overlapping sensory representations, which would allow options to be grouped downstream into categories based on sensory similarity and assigned with common values (*Seger, 2008*). Such a coding-scheme would allow animals to distinguish between options in different categories and perform appropriate behavioral responses (*Kudryavitskaya et al., 2021*). For example, the most

adaptive response when faced with a choice is to pick the highest value stimulus, taking into account the values of available alternatives (*Glimcher and Fehr, 2013*; *Hayden, 2018*; *Padoa-Schioppa and Conen, 2017*). However, this scheme would not readily allow for distinguishing between two options from the same category. It is therefore essential in such a coding-scheme that category boundaries be flexible. One prominent hypothesis suggests that animals directly compare the values of the available stimuli and make use of this relative value signal to guide flexible categorization (*Itti and Koch, 2001*; *Carello and Krauzlis, 2004*; *Mysore and Knudsen, 2011*; *Mysore et al., 2011*). In this study, we ask how this flexibility arises and identify neural correlates of stimulus comparison in the relatively simple learning and memory circuitry of the *Drosophila* mushroom body.

This flexibility can be studied by examining how animals use an associative memory in two different tasks: discrimination and generalization (*Mackintosh, 1974*). In a discrimination task, the animal has to choose between a cue associated with reward and a second cue that could either be similar to (hard discrimination) or distinct (easy discrimination) from the original cue. In the generalization task, the flies have to choose between a cue that is perceptually similar to the trained cue and a cue that is very different. The correct choice in this task depends on the animal generalizing its learned response to the similar cue. So the response to the perceptually similar cue differs between the two tasks – the animal chooses it when generalizing and chooses against it when discriminating. Despite the need to switch choices, performance can be extremely high on both these types of tasks (*Campbell et al., 2013*; *Xu and Südhof, 2013*; *Chen and Gerber, 2014*), suggesting that comparisons between available alternatives have a strong impact on animals' behavioral responses.

We employed these paradigms using aversive olfactory conditioning in *Drosophila*, as its well-studied memory circuit provides a strong framework to understand the neural basis of flexible categorization. Olfactory learning takes place in the mushroom body (MB), where odors are represented by sparse activity patterns of the 2000 intrinsic neurons termed Kenyon cells (KCs) (*Turner et al., 2008*; *Murthy et al., 2008*; *Honegger et al., 2011*). Although different odor response patterns are largely uncorrelated, chemically similar odors can elicit partly overlapping patterns of activity (*Campbell et al., 2013*; *Lin et al., 2014*). These sensory representations are converted into value-representing memory traces that guide behavioral outputs downstream of the KCs, at the synapses that they form with MB output neurons (MBONs) (*Aso et al., 2014b*; *Hige et al., 2015a*; *Owald et al., 2015*; *Villar et al., 2022*). These ~30 distinct MBONs form compartments that integrate input from different subsets of KCs (*Ito et al., 1998*; *Strausfeld et al., 2003*; *Lin et al., 2007*; *Tanaka et al., 2008*; *Aso et al., 2009*; *Aso et al., 2014a*; *Takemura et al., 2017*; *Li et al., 2020*). Furthermore, these synapses are plastic and primarily undergo depression as flies learn an olfactory association (*Berry et al., 2018*; *Cohn et al., 2015*; *Hige et al., 2015a*; *Perisse et al., 2016*; *Séjourné et al., 2011*; but see also *Plaçais et al., 2013*; *Stahl et al., 2022*). This plasticity is mediated by dopaminergic neurons (DANs) that convey information about reward or punishment and arborize in corresponding compartments as the MBONs forming a series of DAN-MBON modules (*Aso et al., 2014a*).

The degree to which KC response patterns overlap has been shown to drive the specificity of learning and resulting behavior (*Campbell et al., 2013*; *Lin et al., 2014*). During learning, synapses from odor-activated KCs to specific MBONs are depressed. A similar odor with an overlapping KC response pattern thus also exhibits a reduced synaptic drive onto the same MBONs (*Hige et al., 2015a*; *Perisse et al., 2016*; *Berry et al., 2018*). The greater the overlap, the more extensive the depression of this other odor's activation of the MBONs, and the greater the generalization. In contrast, when overlap is low, downstream MBON activity is minimally affected and the animal discriminates between the two cues. Although this model has a lot of explanatory power, it does not include a means for explicitly comparing between available options. A comparison would allow for the estimation of a relative value between the available options and explain the high performance of flies in both generalization and discrimination tasks. The explanatory framework must therefore move beyond the idea of overlap in sensory representations.

In this study, we combined neural activity measurements with behavioral experiments, to expand our understanding of flexible categorization. We observed that flies could achieve high levels of performance for both discrimination and generalization tasks and identified a single MB compartment capable of supporting both. Surprisingly, MBON responses in this compartment showed no measurable stimulus-specificity to simple pulses of the two similar odors we used, despite being able to distinguish them behaviorally. However, when we presented odors in sequence, one transitioning

immediately into the other, similar to what flies experience in the behavioral task, we found that MBON responses to these odors were clearly distinct. These findings show that MBON activity is modulated by a temporal comparison of the alternatives presented to the fly, allowing for switches in the categorization of odor stimuli. Importantly, KC representations did not show categorization switching to either simple stimuli or transitions suggesting the involvement of downstream mechanisms. Moreover, behavioral experiments showed that these comparisons are made when stimuli are experienced close together in time. Both imaging and behavior provide complementary evidence that comparing available alternatives 'side-by-side' in time is important for flexible categorization.These results show that the MB circuit implements a comparison, augmenting small differences between overlapping sensory representations to guide flexible stimulus categorization and choice behavior.

## Results

### Precision of memory recall depends on MB compartment

Previous work has shown that flies are capable of high levels of performance on both hard discrimination and generalization tasks (*Campbell et al., 2013*). This study identified a trio of odors to use for experiments on the specificity of memory, based on the degree of overlap of KC response patterns: pentyl acetate (PA) butyl acetate (BA) and ethyl lactate (EL) (*Figure 1A*, left). PA and BA are chemically similar and elicit highly overlapping response patterns in the KC population (*Campbell et al., 2013*). EL is distinct, both chemically and in terms of KC response patterns. Choices between different combinations of these cues can be used to test flies' ability to flexibly classify odors and measure memory specificity. Take, for example, an experiment where flies are trained to form an association with PA. We can present flies with a difficult discrimination task by giving them a choice between the similar odors (PA and BA), or an easy discrimination with a choice between the paired odor (PA) and the dissimilar odor (EL) (*Figure 1A*, right). We can also test whether the association with PA generalizes to the similar odor BA, by giving flies a choice between BA and EL. Since we use these odors in many different combinations for different task structures, with and without reciprocal design, here we will use A to refer to the paired odor (PA or BA) and A' to refer to the other similar odor, which is unpaired, while B always refers to the dissimilar odor, EL. With this nomenclature, hard discrimination involves an A versus A' choice, easy discrimination is A versus B and generalization is A' versus B (*Figure 1A*).

Although previous work showed flies can flexibly categorize odors and learn both generalization and discrimination tasks using these odors, electric shock was used as the reinforcement (*Campbell et al., 2013*). Consequently the synaptic changes responsible were likely distributed across many areas of the mushroom body, and possibly elsewhere. To confine plasticity to a more restricted region of the brain, we used optogenetic reinforcement, pairing the activation of specific DANs with odor presentation (*Figure 1B*; *Claridge-Chang et al., 2009*; *Schroll et al., 2006*). We used drivers to express CSChrimson in specific DANs from the PPL1 cluster that target different compartments involved in aversion learning: α3 (MB630B) and γ2α'1 (MB296B) (*Aso et al., 2014a*; *Aso and Rubin, 2016*). Since compartments have different time courses for memory acquisition and recall (*Aso and Rubin, 2016*), the number of repetitions of odor-reinforcement pairing and the time between training and testing differed depending on the compartment tested (see Methods).

We found that these two compartments exhibited contrasting properties in the easy and hard discrimination tasks (*Figure 1C*). Flies that received reinforcement from DAN PPL1-α3 were poor at the hard discrimination, although they performed significantly better on the easy task (*Figure 1D*, p=0.007, n=12). On the other hand, flies that received optogenetic reinforcement via DAN PPL1-γ2α'1 performed the hard discrimination as effectively as the easy discrimination (*Figure 1E*, p=0.08, n=12). Empty driver controls performed no better than chance at either easy or hard discrimination (*Figure 1—figure supplement 1A*, p=0.052, p=0.38, n=12). These results show that these two compartments have different capacities for discrimination, with α3 weakly discriminating and γ2α'1 stronger.

The difference in ability to support fine discrimination between these two compartments raises the question of whether and how they differ in a generalization task. In a simple model where performance reflects overlap between the test stimulus and the trained odor, the harder the discrimination, the easier the generalization. Does the weakly discriminating α3 compartment support strong

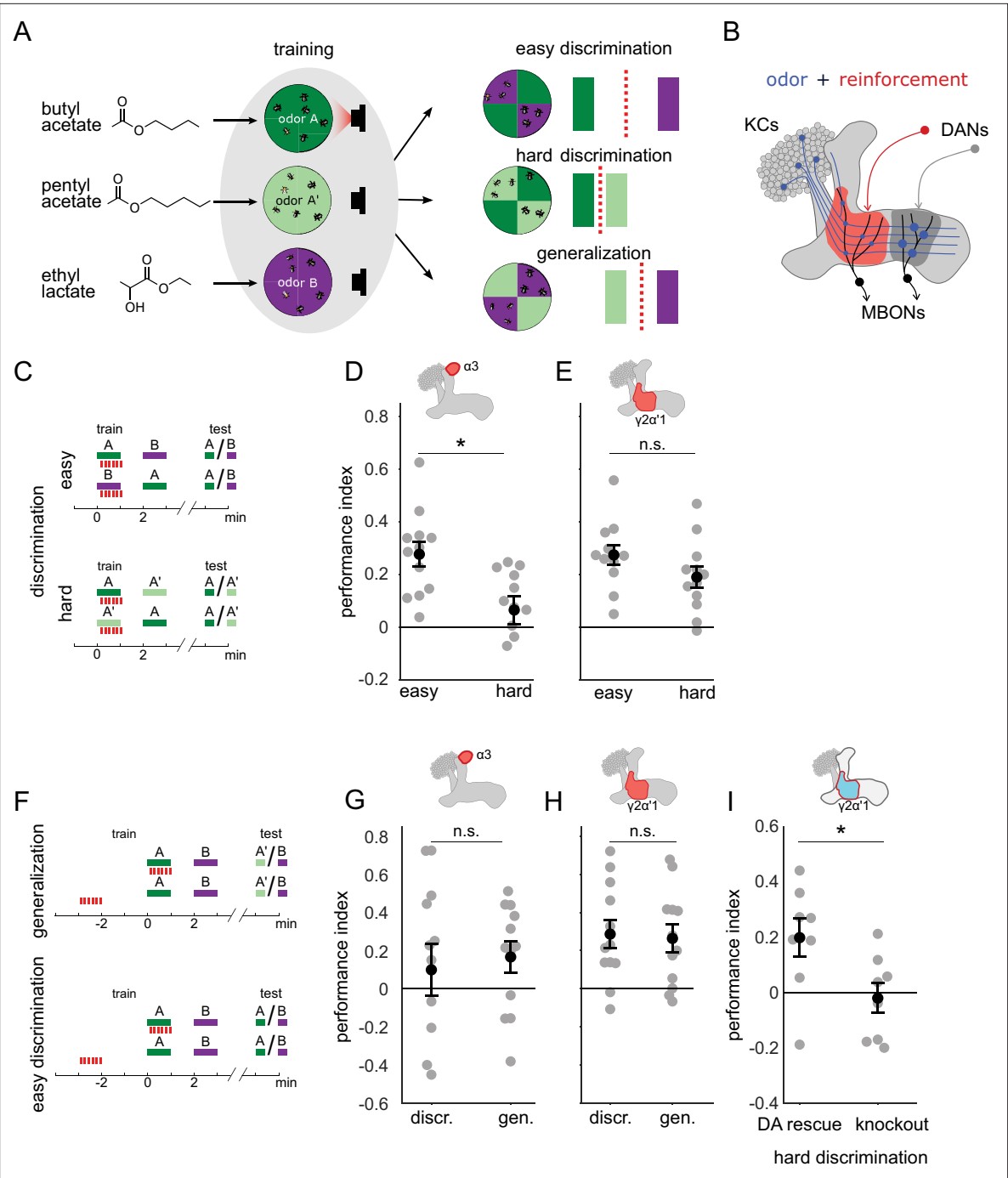

**Figure 1.** A single set of changed synapses can result in generalization or discrimination. (**A**) Left: Chemical structures of the three odors used in the study, the similar odors butyl acetate (BA) and pentyl acetate (PA) and the dissimilar odor ethyl lactate (EL). Middle: During training the similar odors are interchangeably used as the odors that are paired (A) or unpaired (A') with optogenetic reinforcement (LED). Right: Trained flies are then given one of three different choices between odors in opposing arena quadrants. These choices represent the three kinds of tasks used here to study memory specificity. Performance index measures the bias in the distribution of flies across the different quadrants (see Methods). The circles depict fly population behavior in our arenas and the vertical bars depict stimulus choices. The dashed, red line depicts the discrimination boundary in each choice. This boundary shifts relative to the light-green stimulus, depending on the options. (**B**) Mushroom body learning schematic. KCs activated by an odor (blue) form synapses on MBONs in two compartments (red and gray shading). Reinforcement stimulates the DAN projecting to one compartment (red) leading to synaptic depression. (**C**) Behavior protocols for discrimination tasks at two levels of difficulty. Colored bars represent odor delivery periods, red dashes indicate LED stimulation for optogenetic reinforcement. A represents the paired odor, A' the similar odor and B the dissimilar odor. (**D**) Significantly lower performance on the hard discrimination task with reinforcement to α3 (p=0.007, n=12). Flies received 10 cycles of training and were

*Figure 1 continued on next page*

*Figure 1 continued*

tested for memory 24 hours later. CsChrimson-mVenus driven in DAN PPL1-α3 by MB630B-Gal4. (**E**) No significant difference in performance on easy versus hard discrimination with reinforcement to γ2α′1 (p=0.08, n=12 reciprocal experiments). Flies received three cycles of training and were tested for memory immediately after. CsChrimson-mVenus driven in DAN PPL1 γ2α′1 by MB296B-Gal4. (**F**) Behavior protocol for generalization. Scores here are compared to a control protocol where light stimulation is not paired with odor presentation in time. (**G**) No significant difference in performance on generalization and easy discrimination with reinforcement to α3 (p=0.84, n=12). Flies received 10 cycles of training and were tested 24 hr later. (**H**) No significant difference in performance on generalization and easy discrimination with reinforcement to γ2α′1 (p=0.89, n=12 unpaired control performance scores). Flies received three cycles of training and were tested immediately after. (**I**) Rescue of the dopamine biosynthesis pathway in DAN PPL1-γ2α′1 is sufficient for performance on the hard discrimination task (p=0.04, n=8). Black circles and error bars are mean and SEM. Statistical comparisons made with an independent sample Wilcoxon rank sum test.

The online version of this article includes the following figure supplement(s) for figure 1:

**Figure supplement 1.** Control behavior experiments.

performance on generalization, while the strongly discriminating γ2α′1 compartment does not? Or does the γ2α′1 compartment somehow have the flexibility to support strong performance on both tasks?

We tested this by examining relative performance on generalization and easy discrimination tasks in these two compartments. We kept a parallel structure between the two types of tasks by quantifying performance against control experiments where optogenetic stimulation was delivered unpaired to odor delivery (*Figure 1F*; see Methods; note that since these experiments did not have reciprocal controls the performance scores in *Figure 1G–H* are computed differently than in *Figure 1D–E*). As expected, training flies using DAN PPL1-α3 yielded similarly high performance on both generalization and easy discrimination tasks (*Figure 1G*, p=0.84, n=12), while empty driver controls performed no better than chance (*Figure 1—figure supplement 1A and B*, p=0.052, p=0.91, n=12). However, performance on the generalization task was also high in the strongly discriminating compartment γ2α′1, with a performance level indistinguishable from that in the easy discrimination task (*Figure 1H* p=0.89, n=12).

Although the experiments above target optogenetic punishment to specific sites within the MB, there is the possibility that there are secondary sites of plasticity that contribute to the behavioral performance we observe, via indirect connections between MB compartments. To more rigorously confine plasticity to γ2α′1, we performed an experiment where dopamine production is restricted solely to DAN PPL1-γ2α′1 within the fly. Dopamine is necessary for flies to show any measurable aversive learning (*Aso et al., 2019*; *Kim et al., 2007*; *Qin et al., 2012*), and its production requires the *Drosophila* tyrosine hydroxylase enzyme, DTH (*Cichewicz et al., 2017*; *Neckameyer and White, 1993*; *Riemensperger et al., 2011*). So we examined performance of flies lacking DTH throughout the nervous system (*Cichewicz et al., 2017*), but with production rescued specifically in PPL1-γ2α′1 by driving expression of UAS-DTH using the split hemidrivers TH-DBD and 73F07-AD (*Aso et al., 2019*). Performance was significantly higher for the DTH-rescue flies than for the mutants in hard discrimination task (*Figure 1I*, p=0.038, n=8) and generalization tasks (*Figure 1—figure supplement 1D*, p=0.041, n=6), indicating that plasticity in this set of synapses is sufficient for both behaviors (For control experiments with easy discrimination, see *Figure 1—figure supplement 1C*).

These results show that a single memory trace formed via plasticity confined to γ2α′1 supports strong performance on the hard discrimination and generalization tasks. We note that the choice outcomes of these paradigms are opposite: in the generalization experiments flies distribute away from odor A′, while in the hard discrimination task, flies accumulate in the A′ quadrant. We next sought to understand how plasticity in this one compartment can result in this flexible categorization of A′.

## KC inputs to both MB compartments contain enough information for discrimination

We started by evaluating whether the odor inputs to the γ2α′1 and α3 compartments carry enough information to discriminate between the two similar odors used in our behavior experiments. Previous measurements of KC responses to these odors showed that they exhibit overlapping response patterns, but did not determine whether that overlap was differentially distributed across different KC subtypes (*Campbell et al., 2013*). We used two-photon calcium imaging to measure cell population responses in the KC subtypes that send axons to γ2α′1 (γ and α′/β′ KCs) and α3 (α/β KCs) (*Figure 2A*

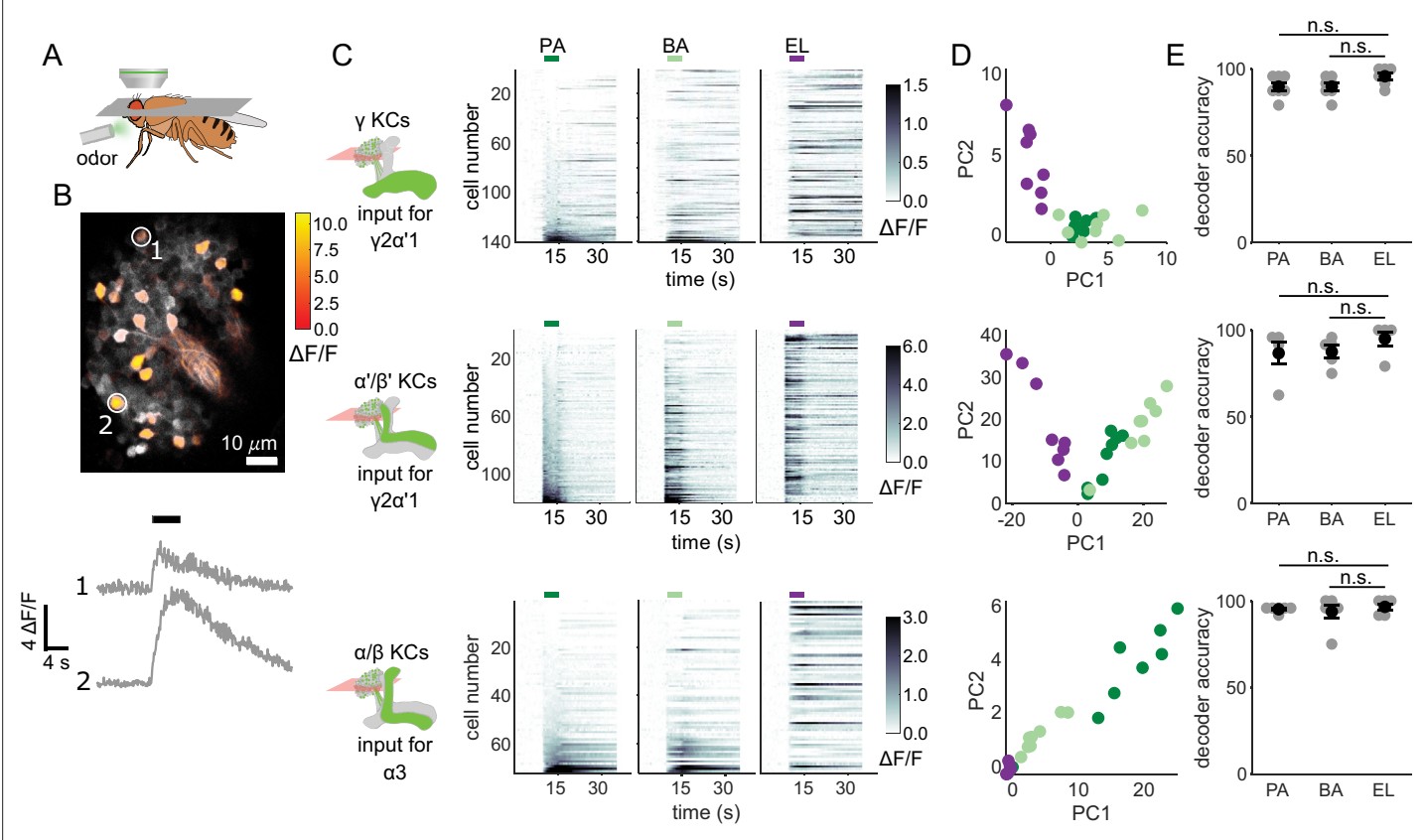

**Figure 2.** KC responses to single odor pulses contain enough information to discriminate similar odors. (**A**) Schematic of in vivo imaging preparation. (**B**) Example single-trial odor response patterns in α/β KCs. ΔF/F responses (color bar) are shown overlaid on baseline fluorescence (grayscale). Numbered circles indicate cells for which ΔF/F traces are plotted below. Black bar indicates odor delivery. (**C**) ΔF/F responses of different KC subtypes to the three odors used in this study. Rows show responses of individual KCs, averaged across trials, sorted by responses to PA. GCaMP6f was driven in γ KCs by d5HT1b-Gal4, in α′/β′ KCs by c305a-Gal4 and in α/β KCs by c739-Gal4. Colored bars above plots indicate the odor delivery period. (**D**) Odor response patterns for the same example flies as in **C**, projected onto the first two principal component axes to show relative distances between representations for the different odors. (**E**) Decoder prediction accuracies, plotted across flies. Each gray circle is the accuracy of the decoder for one fly for a given odor, averaged across all trials. Black circles and error bars are means and SEM. For γ KCs (top), decoder accuracies for PA (n=7 flies, p=0.06) and BA (p=0.08) were not significantly different from EL accuracy. This was also true for α′/β′ KCs (middle, n=5 flies, p=0.13 for the PA-EL comparison and p=0.13 for BA-EL) and α/β KCs (bottom, n=6 flies, p=0.63 for PA-EL and p=0.73 for BA-EL). All statistical testing was done with a paired-sample, Wilcoxon signed rank test with a Bonferroni-Holm correction for multiple comparisons. (**C,D**) In this figure, each shade of green denotes one of the two similar odor chemicals. But in subsequent figures, the darker shade represents the odor paired with reinforcement and the lighter shade, the unpaired, similar odor. In the reciprocal design we use, each of the odor chemicals is the paired odor in half the experimental repeats.

The online version of this article includes the following figure supplement(s) for figure 2:

**Figure supplement 1.** Similar odors have similar KC response patterns.

*and B*). In separate sets of flies, GCaMP6f (*Chen et al., 2013*) was expressed in γ KCs (d5HT1b *Yuan et al., 2006*), α′/β′ KCs (c305a *Armstrong et al., 2006*; *Krashes et al., 2007*) and α/β KCs (c739 *McGuire et al., 2001*). γ and α′/β′ KCs had to be imaged separately since there is no driver that exclusively labels both subtypes. The trial-averaged response traces of individual KCs for each of the three subtypes showed that many of the same cells respond to the two similar odors (PA and BA), but representations did not completely overlap (*Figure 2C*). Responses were very different for the dissimilar odor, EL. KC population response vectors from single trials, plotted as projections along the first two principal component axes (*Figure 2D*), also show the similarity in KC representations between the chemically similar odors. Finally, we examined the similarity of responses for individual KCs to the different pairs of odors. Pooling cells across all imaged flies, we found that similar odors elicited similar response strengths in individual KCs (*Figure 2—figure supplement 1A*, γ KCs: r=0.74, p<0.001; α′/β′ KCs: r=0.76, p<0.001 and α/β KCs: r=0.63, p<0.001). Correlation coefficients were

lower and were not significant for the dissimilar odors (*Figure 2—figure supplement 1B*, γ KCs: r=0.04, p=0.60; α'/β' KCs: r=0.06, p=0.55 and α/β KCs: r=–0.04, p=0.77).

To quantify how effectively KC activity patterns could distinguish between odors, we used logistic regression models to determine the probability a particular odor evoked the KC activity pattern observed on a given trial. We trained logistic regression decoders to recognize KC response patterns using leave-one-out cross-validation. We computed the average decoder accuracy for the 8 odor presentation trials of each odor for each fly. Decoder accuracies for the two similar odors were as high as they were for the dissimilar odor, across all KC subtypes (*Figure 2E*)(γ KCs: comparing accuracies for PA and EL p=0.06, BA-EL p=0.08, n=7 flies; α'/β' KCs: PA-EL p=0.13, BA-EL p=0.13, n=5 flies; α/β KCs: PA-EL p=0.63, BA-EL p=0.73, n=6 flies).

Even though compartments γ2α'1 and α3 receive olfactory input from totally distinct subsets of KCs, input activity patterns appear capable of supporting fine discrimination in all three KC subtypes. Is this information retained one synapse downstream, when hundreds of KCs converge onto the MBONs in these two compartments?

## Plasticity in MBON γ2α'1 is not sufficiently odor-specific for discrimination

We next examined plasticity in the downstream MBONs, to test whether activity of MBON-γ2α'1 could potentially support fine discrimination after training. We carried out on-rig optogenetic reinforcement, and imaged MBON-γ2α'1 odor responses pre- and post-pairing (*Figure 3A*). γ2α'1 spans parts of both the γ and α' MB lobes, but receives reinforcement from the single DAN, PPL1-γ2α'1 (*Aso et al., 2014a*). Two MBONs send dendrites to the same region of neuropil; here we treat them as a single cell type, MBON-γ2α'1, and we imaged from their overlapping dendritic projections (*Figure 3B and C*). We expressed Chrimson88.tdTomato (*Strother et al., 2017*) in the DAN PPL1-γ2α'1 (driven by 82C10-LexA which also drives weak expression in compartments α2 and α3 *Pfeiffer et al., 2013*) and opGCaMP6f selectively in MBON-γ2α'1 (MB077B *Aso et al., 2014a*). We imaged MBON-γ2α'1 responses to pulses of all three odors, before and after pairing one of the similar odors with optogenetic reinforcement (*Figure 3A–C*). We delivered two presentations of each odor stimulus before and after pairing, and imaged only one, to minimize adaptation effects (*Berry et al., 2018*). Based on previous studies, we expected to see depression of the MBON-γ2α'1 response specifically (or at least preferentially) for the reinforced odor (*Berry et al., 2018*; *Cohn et al., 2015*; *Hige et al., 2015a*; *Owald et al., 2015*; *Perisse et al., 2016*; *Séjourné et al., 2011*). However, after pairing, MBON-γ2α'1 responses to A and A' were both strongly depressed (*Figure 3D and E*, p=0.001 for A and p=0.001 for A', n=11 flies). In fact we could not detect a difference in response size between the two, even though only one (A) had been paired with reinforcement (*Figure 3F*, p=0.77). As expected, responses to the dissimilar odor (B) were not affected (p=0.18).

This strong depression of MBON-γ2α'1 responses to both similar odors suggests that downstream of the KCs, A and A' are grouped into the same category. Such a grouping should elicit the same behavior response to both odors and allow for generalization. How then do flies discriminate between them after learning in our hard discrimination task? We postulated that the apparent discrepancy between our behavioral observations and measurements of MBON activity might be because we did not adequately reproduce the fly's sensory experience when it is presented as a choice between two odors.

## MBON responses to odor transitions reflect discrimination behavior

When flies make a choice between two odors in the behavioral arena, they encounter an odor boundary, where the concentration of one odor rapidly drops off and the other rises. To mimic this experience while imaging neural activity on the microscope, we designed an odor delivery system to deliver rapid transitions between odors. We characterized the performance of this odor delivery system with a photo-ionization detector to measure odor concentration changes and an anemometer to measure airflow (*Figure 4—figure supplement 1*).

We then examined how plasticity affects MBON-γ2α'1 responses to these odor transitions. As above, we used single odor pulses for training, to match how flies are trained behaviorally. However, we examined MBON responses to odor transitions pre- and post-pairing, to match how flies experience the choice between odors. These results showed a sharp contrast to our observations with

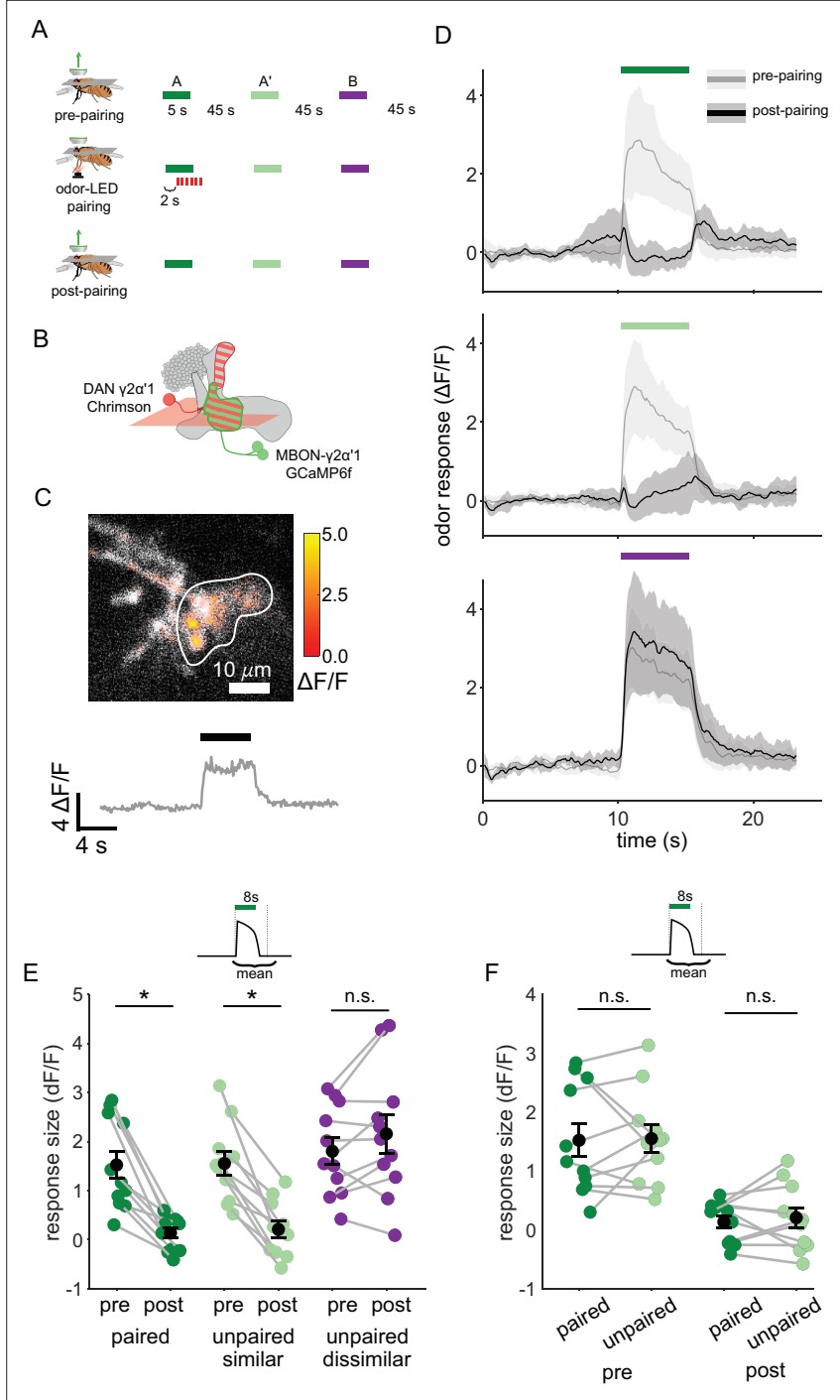

**Figure 3.** Plasticity in MBON γ2α′1 is not sufficiently odor specific for hard discrimination. (**A**) Stimulus protocol for on-rig, in vivo training. Plasticity in MBON γ2α′1 was assessed by imaging pre- and post-pairing with optogenetic reinforcement via DAN PPL1-γ2α′1. There was no imaging during the pairing itself. Colored bars represent 5 s odor delivery (dark green: odor A paired; light green: odor A' unpaired similar; purple: odor B unpaired dissimilar). PA and BA were used as the paired odor for every alternate fly. (**B**) Schematic of experimental design. Expression of GCaMP6f in MBON-γ2α′1 driven by MB077B-Gal4 and Chrimson88-tdTomato in DAN PPL1-γ2α′1 by 82C10-LexA. Imaging plane in the γ lobe as indicated. (**C**) Example MBON-γ2α′1 single-trial odor response. ΔF/F responses (color bar) are shown overlaid on baseline fluorescence (grayscale). White ROI indicates neuropil region for which a single-trial ΔF/F trace is plotted below. Black bar indicates odor delivery. (**D**) MBON γ2α′1 ΔF/F response traces pre- (grey) and post- (black) pairing (mean +- SEM, n=11 flies). Bars indicate 5 s odor delivery period; colors

*Figure 3 continued on next page*

*Figure 3 continued*

correspond to odor identities in a. (**E**) Response sizes pre- and post-pairing show a reduction for both paired (dark green, p=0.001), and similar unpaired (light green, p=0.001) odors but not the dissimilar odor (purple, p=0.18). Response amplitude calculated as mean ΔF/F over an 8 s window starting at odor onset (inset). Connected circles indicate data from individual flies. (**F**) Data as in **E**, re-plotted to compare responses to the paired odor with responses to the unpaired, similar odor before and after training. Response sizes were not significantly different pre- (p=0.77) or post-pairing (p=0.77). Statistical comparisons made with the paired sample Wilcoxon signed rank test, with a Bonferroni Holm correction for multiple comparisons.

single odor pulses. Surprisingly, the depression of responses to A′ seen in single pulses was not readily apparent in an A to A′ transition (*Figure 4A*). Responses were similar to pre-pairing levels when A′ was preceded by A, but not when the order was reversed (*Figure 4B*). Indeed, quantifying the size of the MBON response to the second pulse showed A′ responses were not significantly different pre- and post-pairing (*Figure 4C*; p=0.376, n=13). As expected, responses to A as the second pulse were significantly lower after pairing (*Figure 4C*; p<0.001, n=13). The contrast with the single odor pulse results is clearest when comparing responses to A versus A′ as the second pulse in a transition, where A′ responses were now significantly larger (*Figure 4—figure supplement 2C* p=0.004). Control experiments where LED stimulation was omitted showed no significant differences pre- and post-mock pairing (*Figure 4—figure supplement 3A–C*). Additionally, when we examined responses to A-B and A′-B transitions before and after training, we saw no effect on responses to odor B, indicating that transitions selectively enhance the otherwise depressed responses to the similar odor A′ (*Figure 4—figure supplement 3D–F*).

These results indicate that the way the fly encounters the odor has a profound effect on MBON responses after learning. Isolated pulses of A and A′ elicit similar strongly depressed responses, while transitioning from one odor to the next, as at an odor boundary, responses were clearly distinct, with the A′ response now much stronger. To quantify how effectively MBON-γ2α′1 activity captures an odor boundary, we computed a contrast score reflecting the change in MBON activity at the transition. This score was the difference between the minimum ΔF/F value during the first pulse and the maximum during the second pulse. After learning, contrast around the odor transition was significantly higher for A to A′ transitions than the reverse (*Figure 4—figure supplement 2D*; p=0.001). These results show that the way the animal experiences the odors has a significant effect on how differentially the downstream MBON-γ2α′1 responds to them.

To further evaluate whether this effect contributes to hard discrimination, we next examined the effects of plasticity on odor transition responses in MBON-α3. Reinforcement in this compartment does not support fine discrimination, so if the transition effect is important for discrimination, it should be absent here.

α3 is a slow-learning compartment; when an odor is paired with reinforcement via DAN PPL1-α3, behavioral performance gradually rises until it peaks 24 hr after training (*Aso and Rubin, 2016*). To examine MBON responses when behavioral performance is at this peak, we could not use on-rig optogenetic reinforcement. Instead, we trained flies in a behavior chamber, by pairing an odor with a shock reinforcement. We retrieved flies from the arena and imaged MBON-α3 responses 20–28 hr after training (*Figure 4D*, detailed protocol in *Figure 4—figure supplement 2B*). This experimental approach did not permit us to measure pre- and post-training responses in the same fly. So in these experiments, we compared responses observed in trained flies with those in a mock-trained cohort, where shock was delivered at a different time than odor (*Figure 4—figure supplement 2B*). We note that optogenetic reinforcement was not possible for experiments targeting the α3 compartment; despite extensive efforts we were unable to identify LexA driver lines either with sufficient strength to image MBON-α3 activity, or to get effective reinforcement selectively via DAN PPL1-α3 (*Figure 4—figure supplement 4*).

We found that in response to odor transitions, there was no modulation in MBON-α3 responses (*Figure 4E*). Responses to the second odor in the transition were depressed for both transition orders (*Figure 4F*; A′-A, p<0.001, A-A′, p<0.001, pooled n=14 PA-paired and n=12 BA-paired flies). As expected, responses to the dissimilar odor (B) showed no significant depression (p=0.48). We also evaluated odor boundary detection post-training by computing a contrast score as we did for

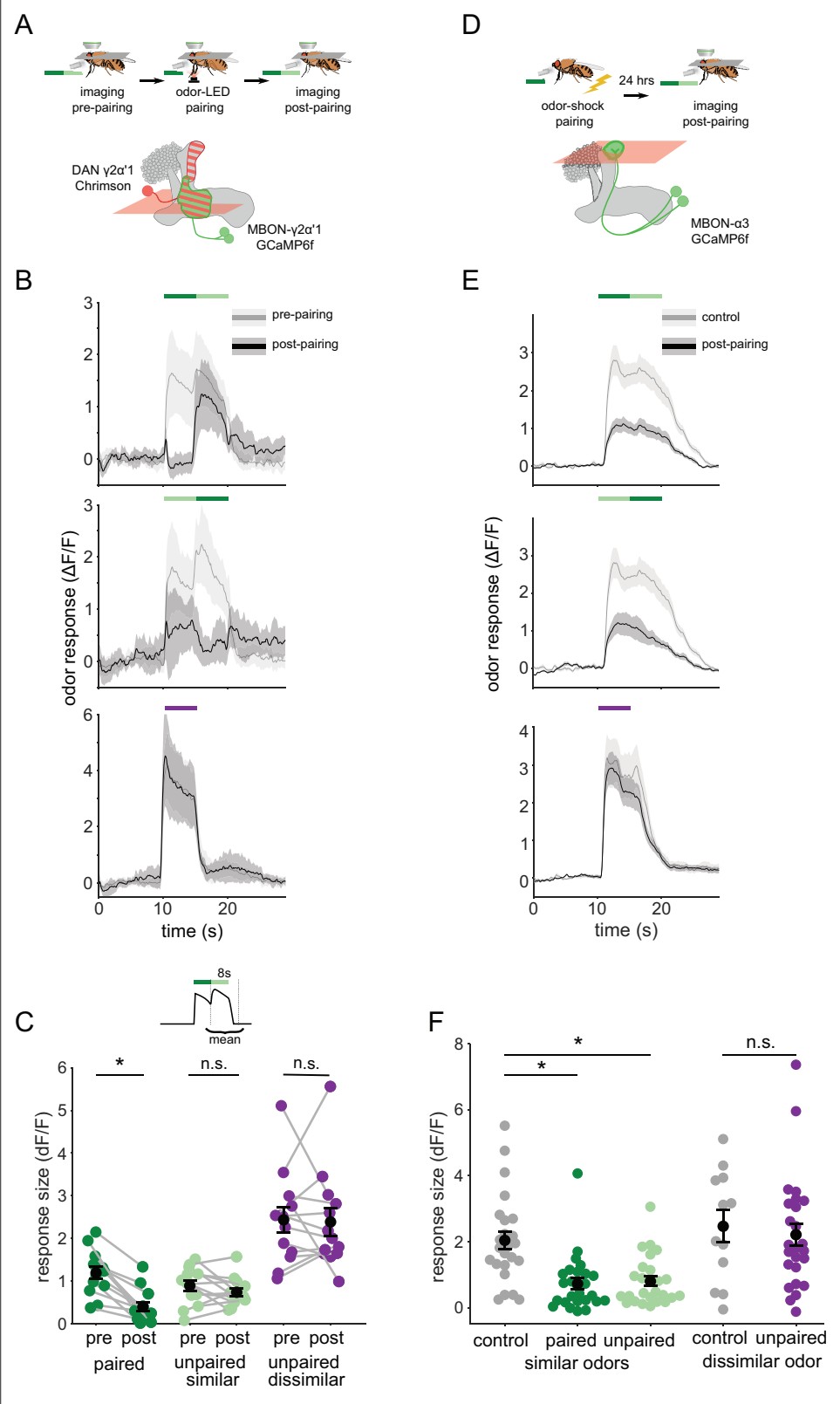

**Figure 4.** Odor transitions enable discrimination by MBON-γ2α′1 but not MBON-α3. (**A**) Schematic of experimental design to assess plasticity in MBON-γ2α′1. Protocol was identical to *Figure 3A*, except odor transitions were used as pre- and post-pairing test stimuli to mimic odor boundaries from the behavioral arena. See *Figure 4—figure supplement 2A* for the detailed protocol. (**B**) MBON-γ2α′1 ΔF/F response time courses pre-

*Figure 4 continued on next page*

*Figure 4 continued*

(gray) and post- (black) pairing to different odor transitions (mean +- SEM, n=13 flies). Colored bars indicate timing of odor delivery, dark green: paired odor, light green: unpaired, similar odor, purple: dissimilar odor. Schematic of the corresponding fly movement in the behavior arena at left. (C) Response sizes in MBON-γ2α′1 pre- and post-pairing for the second odor in the transition. Responses calculated as mean over an 8 s window starting at the onset of the second odor (inset). Responses when the paired odor is second are significantly reduced after pairing (dark green n=13 flies, p<0.001). By contrast there is no significant reduction when the unpaired similar odor comes second (light green p=0.376). The dissimilar odor control showed no significant change (purple, p=1). Connected circles indicate data from individual flies. (D) Schematic of experimental design for MBON-α3. Flies were trained by pairing odor with shock in a conditioning apparatus and odor responses were imaged 24 hr later. Plasticity was assessed by comparing responses against those from a control group of flies exposed to odor and shock but separated by 7 min. See *Figure 4—figure supplement 2B* for the detailed protocol. GCaMP6f was driven in MBON-α3 by MB082C-Gal4 (bottom). (E) MBON-α3 ΔF/F response traces as in **B**. Light gray traces are from control shock-exposed flies (n=12 flies), dark gray traces are from odor-shock paired flies (pooled n=14 PA-paired and n=12 BA-paired flies). Averaged control traces were pooled across trials where either PA or BA was the second odor in a transition. (F) Response sizes in MBON-α3 in control (gray) and trained flies (colored) for the second odor in the transition (computed as in **C**). Responses are significantly reduced in trained flies both when the paired odor is second (dark green, n=14 PA-paired and n=12 BA-paired flies pooled, p<0.001) and when the unpaired similar odor is second (light green, p<0.001). Responses to the dissimilar odor were not significantly different (purple, p=0.48). Plotted control responses were pooled across trials where PA or BA was the second odor in a transition. (**C,F**) Statistical comparisons for MBON-γ2α′1 made with the paired sample, Wilcoxon signed-rank test. For MBON-α3, where responses were compared across different flies, we used the independent samples Wilcoxon's rank-sum test. p-values were Bonferroni-Holm corrected for multiple comparisons.

The online version of this article includes the following figure supplement(s) for figure 4:

**Figure supplement 1.** Time courses of odor delivery and air flow for odor pulses and odor transitions.

**Figure supplement 2.** Contrast around odor transitions is high for MBON-γ2α′1 but not MBON-α3.

**Figure supplement 3.** Observations from no-LED control training and odor transitions to the dissimilar odor.

**Figure supplement 4.** PPL1-α3 split-LexA lines drive expression poorly.

---

MBON-γ2α′1. With MBON-α3, we saw very little contrast at the transition point, and contrast was similarly low for either order of the transition (*Figure 4—figure supplement 2E*, *P*=0.86).

Mimicking the fly's experience in the behavioral arena by presenting odor transitions revealed a strong concordance between neural activity and behavior. In MBON-γ2α′1, when odors are presented in isolation, responses to A and A′ were not measurably different (*Figure 3*). This would allow flies to generalize learning between these odors in most circumstances. But when the two similar odors are juxtaposed in time, matching what they experience when making a choice, MBON-γ2α′1 responses were clearly distinct and could support fine discrimination (*Figure 4*). The ability of MBON-γ2α′1 to respond differently in these two conditions likely reflects the flexible categorization that enables flies to perform both generalization and discrimination. In agreement with this hypothesis, the effect of odor transitions is absent in the α3 compartment, which does not support fine discrimination.

## Odor transition effects on MBONs are not present in the KCs

We have shown that MBON-γ2α′1 responses show a stimulus-history dependent modulation at odor transitions, but MBON-α3 does not. To determine whether this arises upstream of the MBONs, we examined KC responses to odor transitions. Early sensory processing in the antennal lobe could alter odor representations when delivered as transitions, as seen in locusts (*Nizampatnam et al., 2018*; *Saha et al., 2013*). So we examined responses to odor transitions in the input KC populations for MBON-γ2α′1 (γ and α′/β′ KCs) and MBON-α3 (α/β KCs) (*Figure 5A*). We attempted to use the KC activity patterns we measured to reproduce our observations of MBON activity. Specifically, we used logistic regression models, adjusting the weights of KC inputs so that model outputs were low for A and A′ and high for B. To match the training procedure the flies experienced, we first trained the models using isolated odor pulses, and tested predictions for odor transition stimuli. To ensure we did not penalize cells that responded uniquely to transitions, weights were initialized at 1 and we trained models without any weight regularization. Trained weights were negatively correlated with responses to A, as expected (*Figure 5B*, top, Pearson correlation coefficient for γ KCs = - 0.39, α′/β′ = - 0.28 and α/β = - 0.22). These weights were then used to calculate model output for A′-A and A-A′ transitions. In

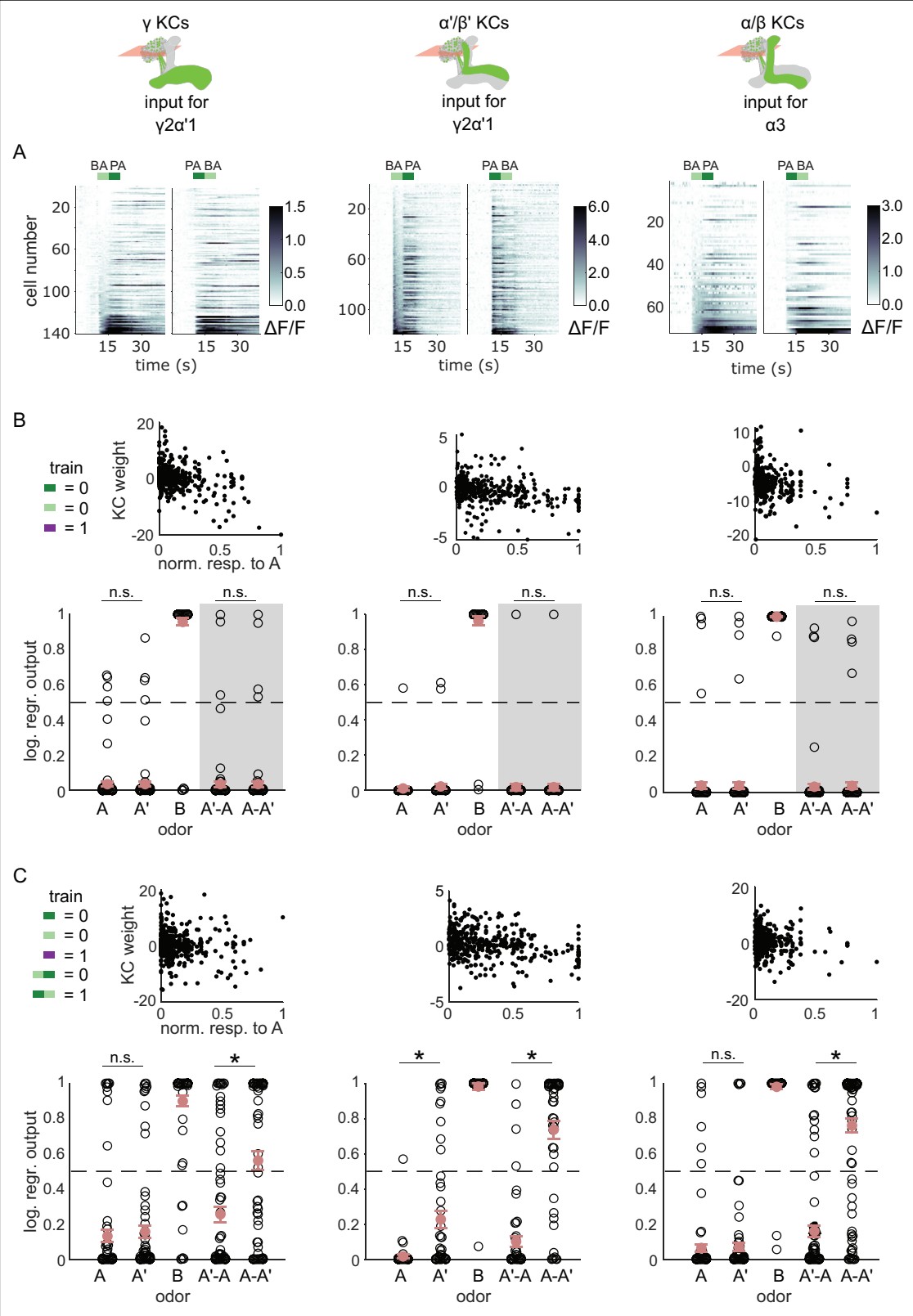

**Figure 5.** KC responses to odor transitions are not sufficient for hard discrimination. (**A**) ΔF/F responses of different KC subtypes to odor transitions. Rows show responses of individual KCs, averaged across trials, sorted by responses to BA-PA. GCaMP6f was driven in γ KCs by d5HT1b-Gal4 (left), in α'/β' KCs by c305a-Gal4 (middle) and in α/β KCs by c739-Gal4 (right). The odor delivery periods are indicated by colored bars at the top. (**B**) We fitted KC weights with logistic regression to give high or low outputs to odors consistent with measured MBON outputs (synaptic weight plots, black circles

*Figure 5 continued on next page*

*Figure 5 continued*

are individual fitted weights, pooled across flies). Individual logistic regression model outputs for held out test data for all types of odor stimuli are plotted in black. The gray background indicates that odor transition data was not part of the training set (n=96 models for γ, n=80 for α'/β' and n=96 for α/β KCs, respectively), red circles and error bars are mean +/-SEM. The dashed, gray line at 0.5 indicates the logistic regression output threshold. Mean model outputs were below the decision threshold for A and A' and were not significantly different (p=1 for γ, p=1 for α'/β' and p=1 for α/β KCs, respectively), as was the case for A'-A and A-A' (p=1 for γ, p=0.95 for α'/β' and p=1 for α/β KCs, respectively). (**C**) Fitted weights and outputs for logistic regression models as in **B**, except that these were trained on single pulse as well as odor transitions. Average model outputs for A and A' were below the decision threshold and were significantly different only for one KC sub-type (p=0.026 for γ, p<0.001 for α'/β' and p=0.062 for α/β KCs, respectively). Mean outputs for A'-A were below the decision threshold, but outputs for A-A' were above it and significantly different for all KC subtypes (p<0.001 for γ, p<0.001 for α'/β' and p<0.001 for α/β KCs, respectively). All statistical comparisons were made with the Wilcoxon signed-rank test with a Bonferroni-Holm correction for multiple comparisons.

The online version of this article includes the following figure supplement(s) for figure 5:

**Figure supplement 1.** KC response patterns are similar for both isolated odor pulses and transitions.

contrast to our observations of MBON activity, model outputs were not significantly different between the two transitions, and were low for both (**Figure 5B**), indicating that transition-evoked changes in the KC odor representations do not underlie the effects on MBON-γ2α'1 activity.

In fact, KC responses to single odor pulses were coarsely similar to responses when those odors came second in a transition; these decoders could also effectively discriminate odors in a transition, although accuracy was slightly lower than with isolated pulses (**Figure 5—figure supplement 1**). To directly evaluate how distinctively the KC population responds to odor transitions, we re-trained the models, adding the requirement that they respond differentially to A-A' versus A'-A transitions. We found that all three KC subtypes could distinguish odor transitions when trained to do so (**Figure 5C**, p<0.001 for all KC subtypes). These results show that it is possible for the model to discriminate transitions, but only if trained using transition-evoked KC activity. By contrast flies learn to discriminate when trained solely with the isolated odor pulses.

Overall, these results show that MBON activity is modulated by a temporal comparison of the alternatives presented to the fly. These observations lead us to the prediction that even if learning is restricted to the γ2α'1 compartment, flies would only be able to discriminate odors if they experienced odor transitions. We tested this prediction with behavioral experiments using odor sequences.

## Odor sequences show that a temporal comparison contributes to odor discrimination

We have shown that MBON-γ2α'1 responses to the similar odors only became distinguishable when presented as transitions. We predicted that flies' behavioral response to these odors should also be indistinguishable, unless they are encountered as transitions. Further, since MBON-γ2α'1 signals positive valence (**Aso et al., 2014b**), our activity measurements predict that flies might be attracted to A' if they encounter an A to A' transition. To test these predictions, we examined behavioral responses to temporal sequences of odor, converting the spatial odor border flies encountered in our earlier behavioral experiments, into an odor transition in time. Flies were trained in the circular arena, and then tested by flooding the entire arena with a sequence of odor pulses. We then compared their behavioral response to direct odor transitions to their response when we interrupted the transition with 25 s of clean air. We determined the timing of odor pulse transitions using photo-ionization detector measurements at the exhaust outlet of the arena (**Figure 4—figure supplement 1**), and analyzed fly behavior around these timepoints.

Attraction to an odor was quantified by how much the flies move upwind; in the arena odors flow inwards from the periphery so we measured displacement away from the center of the arena. We examined the time course of upwind displacement for direct and interrupted transitions (**Figure 6A–E**). We observed strong upwind displacement during the second pulse of an A-A' transition, which was significantly larger than during the reverse A'-A sequence (**Figure 6B, C and F**, p=0.004, n=12). This contrasted with results observed with a 25 s gap in between the two odor pulses. In these interrupted transitions, responses to the second pulse were not significantly different depending on transition order (**Figure 6 D, E and G**, p=0.85, n=12 for A-gap-A', n=13 for A'-gap-A,), and showed a similar degree of upwind displacement to that evoked during the first pulse, as expected. Note that starting locations at the onset of the second odor pulse were not significantly different in any

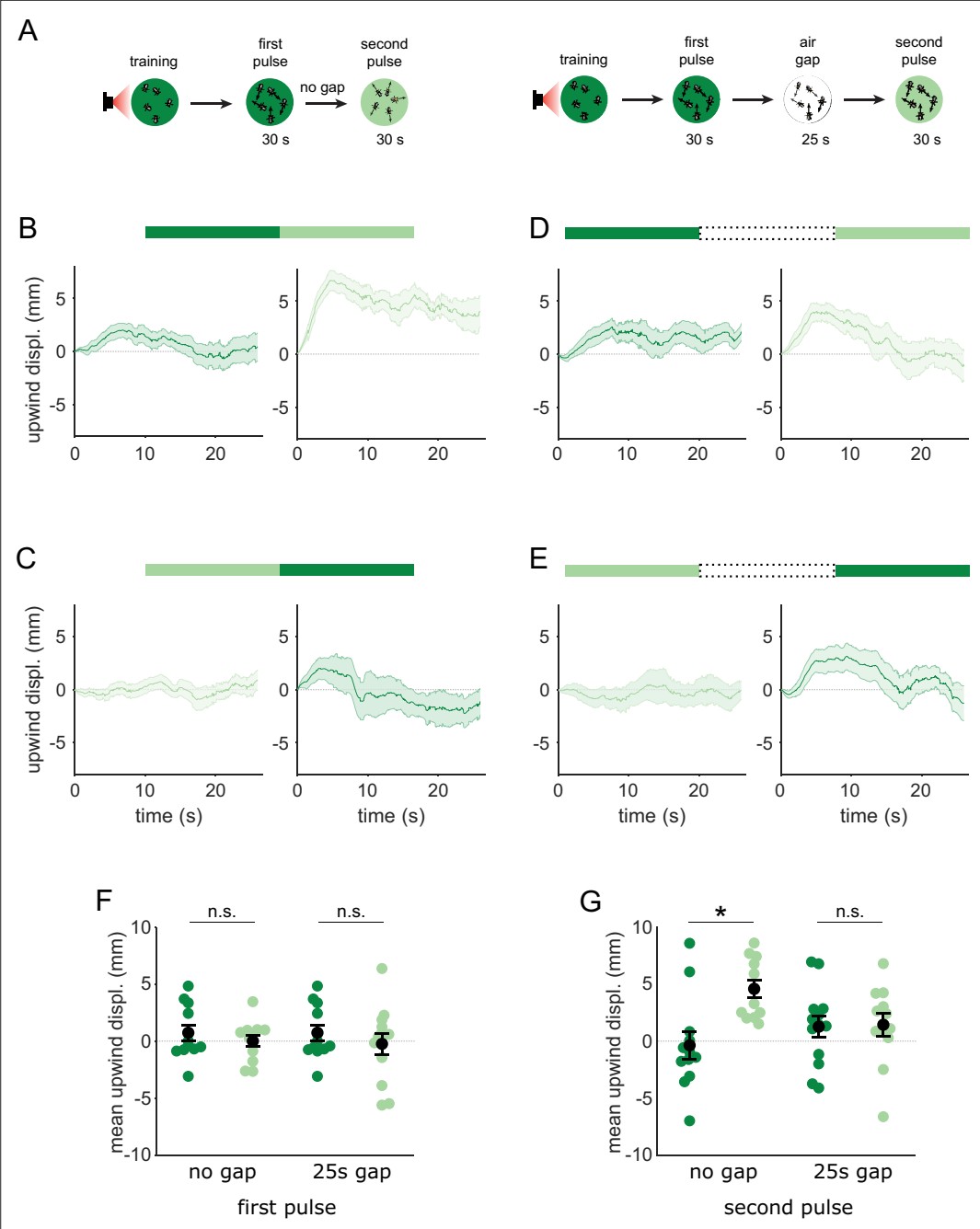

**Figure 6.** Flies are attracted to the unpaired odor only in transitions. (**A**) Experimental strategy for measuring behavioral responses to odor transitions. Flies were trained by pairing one of the similar odors with optogenetic activation of DAN PPL1-γ2α′1. They were then tested with 30 s odor pulses presented either as direct transitions (left) or interrupted by a 25 s air period (right). Schematics illustrate A-A′ transitions but both sequences were tested, as indicated by the bars on top of panels B-E. (**B**) Upwind displacement during the first and second pulses of an A-A′ odor transition, as indicated by the green bars up top. This was computed as the increase in each fly's distance from the arena center over the odor delivery period, then averaged across all flies in an arena (approximately 15 flies per arena). Traces in dark and light green are responses to A and A′ respectively. Plots are mean +/-SEM (n=12 arena runs for all stimulus types). (**C**) Upwind displacement for the reverse odor transition i.e. A′-A. (**D**) Upwind displacement for A-gap-A′ interrupted transition. (**E**) Upwind displacement for the reverse A′-gap-A interrupted transition. (**F**) Upwind displacement in response to the first odor pulse, averaged across flies in each arena experiment. Mean displacement was not significantly different between unpaired and paired odors for experiments with no gap (n=12, 12 experiments for paired and unpaired odors, p=0.70) and a 25 s gap (n=12,

*Figure 6 continued on next page*

*Figure 6 continued*

13, p=0.40). (**G**) As in **F** except for responses to the second odor pulse. Displacement was significantly different for transitions with no gap between pulses (n=12, 12 experiments for paired and unpaired odors, p=0.004) but not different when transitions were interrupted by a 25 s gap (n=12, 13, p=0.85). Statistical comparisons in **F** and **G** were made with the independent-sample Wilcoxon rank sum test with a Bonferroni-Holm correction for multiple comparisons.

The online version of this article includes the following figure supplement(s) for figure 6:

**Figure supplement 1.** Transition dependent attraction is not a result of linearly summed, single-pulse responses.

**Figure supplement 2.** Flies reinforced via DAN PPL1-α3 do not respond to transitions between A and A′.

condition, ruling out the possibility that flies go more upwind with the A-A′ transition because they start from further downwind in the arena (*Figure 6—figure supplement 1B*, n=12 for A-A′, n=12 for A′-A, p=0.08 for direct; n=12 for A-gap-A′, n=13 for A′-gap-A, p=0.39 for interrupted). Additionally, we ruled out the possibility that the increased upwind displacement during such a transition comes from a linear combination of the response to the end of the first odor pulse and the beginning of the second (*Figure 6—figure supplement 1C-G*). These results show that behavioral responses to A′ are distinct only when it immediately follows the paired odor A, matching the odor transition responses we observed in MBON-γ2α′1.

The upwind displacement during the A-A′ transition is consistent with our observation that MBON-γ2α′1, a positive valence MBON that drives upwind behavior, is highly active during these transitions. In fact, the mean upwind displacement after an A-A′ transition was similar to that caused by optogenetic activation of MBON-γ2α′1 in the arena (unpublished communication - Y. Aso). Overall, these results show that flies compare available alternatives 'side-by-side' in time and that stimulus history is important for flexible categorization and behavior. When the two odors are encountered separately, both MBON-γ2α′1 output and fly behavior are indistinguishable for the two similar odors. However, when they are closely apposed in time, MBON-γ2α′1 activity is enhanced and flies are attracted to A′.

## Discussion

Using a pair of perceptually similar odors (A and A′) and one distinct odor (B), we identified a site in the MB circuit that switches the neuronal and behavioral categorization of A′ depending on whether flies are presented an A′ vs A or an A′ vs B choice. Learning-related synaptic plasticity resulted in depressed neuronal responses to both similar odors when presented in isolation, consistent with strong behavioral generalization. However, when the odors were presented sequentially, as at an odor boundary, neuronal responses to A and A′ were distinct. Moreover, behavioral experiments with carefully timed odor delivery showed that flies' response to A′ was distinct from A only if they were delivered in a transition. These results demonstrate how presenting cues as a choice can influence behavioral responses. An odor boundary presents an opportunity to compare stimuli, and this comparison modulates memory traces by amplifying small differences between stimuli to change categorization and behavior.

### Memory specificity is determined by more than overlap of KC somatic activity patterns

KC representations are conventionally thought to be the key player in determining whether to discriminate or generalize. KC activity patterns in response to distinct odors have little overlap (*Perez-Orive et al., 2002*; *Murthy et al., 2008*; *Turner et al., 2008*; *Honegger et al., 2011*; *Campbell et al., 2013*), and this allows synaptic changes to be highly stimulus specific (*Hige et al., 2015a*). However, activity patterns are not so sparse that pattern separation is complete (*Campbell et al., 2013*; *Dasgupta et al., 2017*; *Endo et al., 2020*; *Hige et al., 2015b*). Overlap between different odor response patterns exists, and correlates with both the strength of generalization and the specificity of plastic changes in MBONs. Importantly, prior work did not examine whether the extent of overlap is distributed differentially across different KC subtypes. This could serve as the basis for differences in discrimination we observe between compartments, with optogenetic training in γ2α′1 but not α3

capable of supporting hard discrimination. However, we found that all three major subtypes of KCs exhibit similar levels of overlap across our odor test set. Furthermore, we found that odor representations in all three KC subtypes contain enough information for A-A' discrimination. Despite this, when we examined KC responses to odor transitions, we found that they could not account for the odor-transition responses we observed in MBON-γ2α'1.

Another factor that could potentially contribute to the different specificity in these compartments are the numbers of KC inputs. Theoretical work suggests that – holding the absolute number of responding KCs constant – the larger the total population of KCs, the less overlap there will be between different odor response patterns (*Marr, 1969*; *Albus, 1971*; *Babadi and Sompolinsky, 2014*; *Cayco-Gajic and Silver, 2019*). However, contrary to theoretical expectations, of the two MBON types we studied MBON-α3 received more synaptic inputs from a greater number of KCs, however it was poorer at hard discrimination (MBON γ2α'1=2,959 synapses from 336 α'/β' KCs and 3773 synapses from 683 γ KCs per hemisphere; MBON α3=11,360 synapses from 888 α/β KCs per hemisphere; *Clements et al., 2020*; *Li et al., 2020*), again indicating that compartments' different capabilities for hard discrimination were not a result of differences in sensory representations. We note that our KC activity measurements are all from cell bodies, while KC output synapses are the site of plasticity (*Bilz et al., 2020*), and we cannot formally exclude that activity at KC synapses may differ based on type-specific integrative properties (*DasGupta et al., 2014*; *Groschner et al., 2018*; *Vrontou et al., 2021*), or axo-axonic connections between KCs (*Bielopolski et al., 2019*; *Manoim et al., 2022*).

## Adapting memory usage through stimulus comparisons

Our work instead highlights the importance of comparisons in determining how a stimulus is categorized. In particular we observe that flies make a temporal comparison of inputs and identify the underlying neural implementation of comparison in the MB. Prior work has shown history-dependent effects at multiple layers of the olfactory circuit. For example, in the rodent olfactory bulb, responses to odor sequences are linear combinations of the responses to individual pulses (*Gupta et al., 2015*). On the other hand, in locusts, presenting odors singly or in transitions altered odor representations non-linearly in KCs (*Broome et al., 2006*). The extent of response alteration correlated with the accuracy of behavioral recall (*Saha et al., 2013*). In *Drosophila*, odor transitions can cause similar changes in PN representations that result in altered innate odor preference (*Badel et al., 2016*). In another locust study (*Nizampatnam et al., 2018*), presenting an odor in a transition altered its representation to enhance contrast in the locust antennal lobe glomeruli. These observations suggest that changes in response to transitioning stimuli are mediated by a mechanism that takes place early on in the olfactory circuit. However, our observations of KC activity indicate that, although KCs response patterns to an odor presented as a single pulse versus in a transition are distinct, they are not sufficiently so to generate the sequence-specific transition effect we observe here.

We suggest instead that flexible categorization in *Drosophila* involves a mechanism at or downstream of the KC-MBON synapses modified during learning. Examples of such downstream modulation of memory have already been observed in the MB. In *Drosophila*, recalling food reward associations from one MB compartment is gated by another MB compartment depending on whether or not the fly is hungry (*Perisse et al., 2016*). In this case, recall is regulated by the addition of a layer of contextual modulation through neuropeptide signaling that couples neural activity to satiety state. Ongoing motor activity can also affect dopaminergic inputs along the MB lobes (*Cohn et al., 2015*). These could modulate memory-traces on relatively short timescales based on the behavioral state of the animal. Switching between independently stored short and long-term memories provides another solution (*Trannoy et al., 2011*; *Huetteroth et al., 2015*; *Yamagata et al., 2015*). Experiments have shown that long-term memory allows for more generalization than short-term memory (*Ichinose et al., 2015*; *König et al., 2017*). However, all these mechanisms rely on an *internal* state signal (satiety or locomotion) rather than comparisons between *external* stimuli.

Here we establish a few important constraints on a possible mechanism for flexible categorization in *Drosophila*: (i) it manifests at or downstream of the sites of learning, the KC >MBON synapses and (ii) it modifies responses to stimuli asymmetrically - in A to A' transitions and not the reverse. One mechanism that could satisfy these criteria, would involve an explicit comparison of MBON activities in time, much like the delay-lines in the auditory pathways of owls and crickets (*Schöneich et al., 2015*; *Sullivan and Konishi, 1986*). This could be implemented via a downstream neuron that receives a

real-time and a delayed copy of MBON-γ2α′1 activity, which then provides a positive feed-back signal to the MBON to amplify small increases in activity. Both of these motifs have been observed in the EM connectome (*Li et al., 2020*). As the extent of depression of KC >MBON synapses is inevitably slightly weaker for any odor that overlaps imperfectly with the learned odor, this mechanism would sensitize the circuit to small differences in MBON activity that arise around an odor transition. Another class of mechanisms centers on the observation that the KC population exhibits a distinct pattern of responses to odor offset (*Tanaka et al., 2008*; *Lüdke et al., 2018*). Offset responses in other MBONs can be potentiated (*Vrontou et al., 2021*), presumably due to the timing of reinforcement (*Cohn et al., 2015*; *Handler et al., 2019*), suggesting a similar mechanism might operate in MBON-γ2α′1 to augment responses to the second odor in a transition. An additional candidate mechanism is plasticity of inhibitory input to KCs from the APL neuron. Activity of this inhibitory neuron is reduced by training (*Zhou et al., 2019*; *Liu and Davis, 2009*), so when a non-overlapping set of KCs is activated at an odor transition, that excitation may more effectively drive the downstream MBON. Inhibition at odor offset is a particularly prominent feature of the α′/β′ KCs that are input to MBON-γ2α′1 (*Inada et al., 2017*), so this effect could act in combination with potentiation of KC offset responses to create pronounced changes in KC output at an odor transition. Future work will be needed to resolve these different possibilities.

Many animals use a stored memory to support different behaviors based on the choices available to them. We have shown that in *Drosophila*, this response flexibility relies on comparing cues side by side in time. Making fine-grained distinctions is easier when a temporal comparison is possible, but when it is not, more generalized categorizations can be an adaptive default.

## Methods
### Fly strains
*Drosophila melanogaster* were raised on standard cornmeal food at 21 °C at 60% relative humidity on standard cornmeal food on a 12–12 hr light-dark cycle. For optogenetics behavior experiments, crosses were set on food supplemented with 0.2 mM all-trans-retinal and moved to 0.4 mM after eclosion and kept in the dark throughout.

| Transgene | Expression target/reporter description | Bloomington stock number, reference |
|---|---|---|
| MB296B split Gal4 | DAN PPL1-γ2α′1 | BDSC:68253 *Aso and Rubin, 2016* |
| MB630B split Gal4 | DAN PPL1-α3 | BDSC:68290 *Aso and Rubin, 2016* |
| d5HT1b-Gal4 | γ KCs | BDSC:27637 *Yuan et al., 2006* |
| c305a-Gal4 | α′/β′ KCs | BDSC:30829 *Krashes et al., 2007* |
| c739-Gal4 | α/β KCs | BDSC:7362 *McGuire et al., 2001* |
| MB077B split Gal4 | MBONs γ2α′1 | BDSC:68283 *Aso et al., 2014a* |
| MB082C split Gal4 | MBONs α3 | BDSC:68286 *Aso et al., 2014a* |
| R82C10-LexA | DANs PPL1-γ2α′1, α2, α3 | BDSC:54981 *Pfeiffer et al., 2013* |
| 20XUAS-CsChrimson-mVenus attp18 | Optogenetic activation for behavior | BDSC:55134 *Klapoetke et al., 2014* |
| 13XLexAop2-IVS-Syn21-Chrimson88-tdT-3.1-P10 | Optogenetic activation for imaging | BDSC: n.a. *Strother et al., 2017* |
| 20XUAS-IVS-Syn21-opGCaMP6f-P10 | Codon-optimized Ca$^{2+}$ reporter | BDSC: n.a. *Chen et al., 2013* |

Expression patterns of split-GAL4 lines produced by Janelia FlyLight (*Jenett et al., 2012*) can be viewed online (http://splitgal4.janelia.org/cgi-bin/splitgal4.cgi).

## Behavior

DAN driver split Gal4 crossed with 20XUAS-CsChrimson-mVenus attp18

**TH-rescue experiment** (genetic strategy as in *Aso et al., 2019*)

**knockout**

w, 20XUAS-CSChrimson-mVenus attP18; +; ple², DTHFS ±BAC attP2, TH-ZpGAL4DBD VK00027 / TM6 B

crossed with

w; R73F07-p65ADZp attP40 /CyO; ple², DTHFS ±BAC attP2 /TM6B

**knockout and rescue in DAN PPL1-γ2α′1**

w, 20XUAS-CSChrimson-mVenus attP18; UAS-DTH1m; ple², DTHFS ±BAC attP2, TH-ZpGAL4DBD VK00027 /TM6 B

crossed with

w; R73F07-p65ADZp attP40 /CyO; ple², DTHFS ±BAC attP2 /TM6B

## KC imaging

γ KCs: w; +/+; d5HT1b-Gal4/20XUAS-IVS-Syn21-opGCaMP6f-P10 VK00005

α′/β′ KCs: w; c305a-Gal4/+; 20XUAS-IVS-Syn21-opGCaMP6f-P10 VK00005/+

α/β KCs: w; c739-Gal4/+; 20XUAS-IVS-Syn21-opGCaMP6f-P10 VK00005/+

## MBON γ2α′1 imaging

20XUAS-IVS-Syn21-opGCaMP6f-P10 Su(Hw)attP8 /w; R25D01-ZpGAL4DBD attP40 /82C10-LexAp65 attP40; R19F09-p65ADZp attP2 /13XLexAop2-IVS-Syn21-Chrimson88::tdT-3.1-p10 in VK00005

R25D01 and R19F09 are components of the MB077B stable split-GAL4 driver (BDSC: 68283)

## MBON α3 imaging

w; +/+; 20XUAS-IVS-Syn21-opGCaMP6f-P10 VK00005 /R23C06-ZpGAL4DBD in attP2, R40B08-p65ADZp VK00027

R23C06 and R40B08 are components of the MB082C stable split-GAL4 driver (BDSC: 68286)

## Behavior experiments

### Odor quadrant choice assay

Groups of approximately 20 females, aged 4–10 d post-eclosion were anaesthetized on a cold plate and collected at least two day prior to experiments. After a day of recovery on 0.4 mM all-trans-retinal food, they were transferred to starvation vials containing nutrient-free agarose. Starved females were trained and tested at 25 °C at 50% relative humidity in a dark circular arena described in *Aso and Rubin, 2016*. The arena consisted of a circular chamber surrounded by four odor delivery ports that divide the chamber into quadrants. The input flow rate through each port was 100 mL/min, which was actively vented out a central exhaust at 400 mL/min. Odors were pentyl acetate, butyl acetate and ethyl lactate (Sigma-Aldrich product numbers 109584, 287725, and W244015 respectively). Except for the TH-rescue experiments shown in *Figure 1I*, these odors were diluted 1:10000 in paraffin oil (Sigma-Aldrich product number 18512). For the experiments in *Figure 1I*, we used a different odor delivery system which utilizes air dilution of saturated odorant vapor, and delivered odors at a 1:16 dilution of saturated vapor.

Flies were aspirated into the arena via a small port, and allowed 60 s to acclimatize before training commenced. Training consisted of exposing the flies to one of the odors while providing optogenetic stimulation via a square array of red LEDs (617 nm peak emission, Red-Orange LUXEON Rebel LED, 122 lm at 700mA) which shone through an acrylic diffuser to illuminate flies from below. LED activation consisted of 30 pulses of 1 s duration with a 1 s inter-flash interval, commencing 5 s after switching on the odor valves and terminating 5 s after valve shut-off.

To optimize learning scores, we used different training regimes depending on the compartments receiving optogenetic reinforcement, according to *Aso and Rubin, 2016*. A single training session was used for MB296B, TH-mutant, TH-rescue, while 3 training sessions, separated by 60 s, were used for some MB296B experiments, as indicated in the text. For MB630B we used 10 training sessions separated by 15 min.

Following training, testing was carried out with the appropriate odors for each task. In the test configuration, the two different odor choices are presented in opposing quadrants for 60 s. Videos of fly behavior were captured at 30 frames per second using MATLAB (Mathworks, USA) and BIAS (http://archive.iorodeo.com/content/basic-image-acquisition-software-bias.html) and analyzed using custom-written code in MATLAB.

### Odor attraction assay
For the odor attraction assay, the outputs of odor machines were re-configured to inject the output of a single odor machine into all four quadrants. We switched output from one machine to the other to deliver rapid odor transitions in time. About 15 flies were introduced into the arena for each experiment. The rest of the behavioral procedures were identical to those used in the quadrant choice assay.

### Optogenetic MBON-activation assay
For this assay, a clean air stream was delivered into all four arena quadrants throughout the experiment. Flies expressed CSChrimson in MBON γ2α'1. Flies received six 10 s long LED flashes, separated by 60 s of darkness. The rest of the behavioral procedures were identical to those used in the quadrant choice assay.

## Calcium imaging
Flies were imaged on a resonant-scanning, Janelia, jET MIMMS2.0 custom-designed two-photon microscope, with a Chameleon Ultra II, Titanium-sapphire laser (Coherent, USA) tuned to emit 920 nm. Images were acquired using a 20 x, NA 1.0, water-immersion objective lens XLUMPLFLN (Olympus, Japan) and a GaAsP PMT H11706P-40 SEL (Hamamatsu, Japan). Power after the objective ranged from 4 to 5 mW for MBON imaging and 4–7 mW for KC imaging, depending on the preparation. Microscope control and data acquisition ran on the Scanimage platform (Vidrio, USA). Frames were acquired at 30 Hz, but three frames at a time were averaged during acquisition, for a final frame rate of 10 Hz. For KC imaging, pixels were sampled at 0.22 μm/pixel and for MBON imaging, at 0.18 μm/pixel. For photostimulation, flies were fully illuminated from beneath with 617 nm light through a liquid light-guide (LLG-03-59-340-0800-2, Mightex, USA) butt-coupled to an LED light source (GCS-0617–04 A0510, Mightex, USA). Intensity at the fly was 1 mW/mm$^2$. LED pulses were delivered at a frequency of 1 Hz, with a duty-cycle of 50%, for 5 s, starting 2 s after paired-odor onset.

For optogenetics imaging experiments, crosses were set on food supplemented with 0.4 mM all-trans-retinal, and maintained on the same food at 25 ° C until flies were used for experiments. Flies were prepared as described previously (*Campbell et al., 2013*; *Honegger et al., 2011*). Three- to 8-day-old female flies were immobilized in a 0.25 mm thick stainless-steel sheet with a photo-chemically etched tear-drop shaped hole (PhotoFab, UK) and glued into place with two-component epoxy (Devcon, USA). For imaging in the KC somata and the MBON dendrites, head angle was adjusted differently to give best optical access to the target region, taking care to keep the antennae dry beneath the metal plate. For KC and MBON α3 imaging, the back of the head was submerged in Ringer's bath solution consisting in mM: NaCl, 103; KCl, 3; CaCl2, 1.5; MgCl2, 4; NaHCO3, 26; N-tris(hydroxymethyl) methyl-2-aminoethane-sulfonic acid, 5; NaH2PO4, 1; trehalose, 10; glucose, 10 (pH 7.3, 275 mOsm). For γ2α'1 MBON experiments, the flies were starved (24 hours in nutrient-free, distilled-water agarose vials). Previous studies have shown that hemolymph sugar is halved in flies starved for 24 hrs (*Dus et al., 2011*). So we used bath Ringer's where glucose and trehalose were halved to 5 mM each and the non-metabolizable sugar arabinose (10 mM) was substituted to maintain osmolarity. For KC imaging, once the brain was exposed, bath solution was momentarily aspirated away and the preparation was covered in a drop of 5% (w/v) agarose (Cambrex Nusieve, catalog #50080) in Ringer's, cooled to 36 ° C, which was then flattened with a 5 mm diameter circular coverslip that was then removed just prior to imaging.

For KC imaging, 8 repeats of single odor pulses and each kind of odor transition were delivered with an inter trial interval of 45 s. Stimulus types were randomly interleaved. For MBON γ2α'1 imaging, we delivered two repeats of either single odor pulses or transitions before and after odor-reinforcement pairing (*Figure 4—figure supplement 4A*), adapted from *Berry et al., 2018*. Only one repeat was imaged before and after pairing. For MBON α3 imaging, only the second presentation of each transition stimulus type was imaged (*Figure 4—figure supplement 4B*).

## Odor delivery for imaging experiments

To deliver rapid odor transitions, we set up two separate odor delivery machines (*Honegger et al., 2011*) and joined their outputs upstream of the final tube delivering odor to the fly. These systems use saturated odor vapor which is then serially diluted in clean air to a final dilution of 0.8% (v/v). This was delivered to the fly at a flow rate of 400 mL/min from a tube with an inner diameter of 3 mm.

We measured relative odor concentrations with a photoionization detector (200B miniPID, Aurora Scientific, Canada). Different chemical vapors at the same concentration generate different PID signal amplitudes. Thus, the PID signal is linearly related to concentration only for a given odor chemical. The PID probe was used to measure and tune odor pulse shapes and to measure and account for the time taken for an odor pulse to reach the fly. The short period of overlap between the fall of the first odor pulse and the rise of the second occurred for both kinds of transition stimuli, paired to unpaired transitions and unpaired to paired transitions (*Figure 4—figure supplement 1C*). Hence, this overlap would not affect our measures of discriminability between the two similar odors. A hot-wire anemometer S490 (Kurz, USA) was used to measure air-velocity at a sampling rate of 10 KHz while mock-odor pulses were being delivered though empty odor-vials. This was to rule out any mechanical transients at the time of odor-transitions being an external cue to the flies. To minimize transients, we combined the steps of the second serial dilution of the second odor pulse and mixing the outputs of the two odor machines. The second pulse in all transition stimuli was introduced into the final air stream at one-tenth the flow-rate. Any pressure-transients due to valve switches during the transition to the second pulse were too small to be measured by the anemometer in the final output. We saw a small valve-switching transient at the beginning of the first pulse in any transition (8% the size of the steady-state flow, *Figure 4—figure supplement 1D*). Since this transient was always at the onset of the first pulse, and not during transitions, again, it did not affect discriminability.

Final air flow rate and odor concentration were adjusted to best match the odor flux a fly would experience in the arena. Since odor flows inward from the circumference of the arena, odor flux increases with distance from the periphery. We computed the odor flux on the circle that covers half the arena area and matched the flux delivered on the rig to it.

## Data analysis

### Behavior

Videos recorded during the test phase were analyzed using custom-written MATLAB code. The centroid of each fly was identified and the number of centroids in each quadrant computed for every frame of the experiment.

For discrimination experiments, a Performance Index (PI) was calculated as the number of flies in the quadrants containing the paired odor minus the number in the quadrants with the unpaired odor, divided by the total number of flies (*Tully and Quinn, 1985*). This value was calculated for every frame of the movie, and the values over the final 30 s of the test period averaged to compute a single PI. Discrimination experiments employed a reciprocal design where the identity of the paired and unpaired odors was swapped and a single data point represents the averaged PI from two reciprocally trained groups of flies.

Generalization experiments could not employ a reciprocal design, so instead we compared scores against control experiments where flies were exposed to LED stimulation that was not paired with odor delivery; instead stimulation preceded odor by 2 min. In this case the PI score reported as a single data point is the PI observed from the generalization experiment minus the PI observed in the unpaired control, after both PIs were corrected for biases in initial quadrant occupancies by subtracting away the pre-odor baseline.

Statistical testing was done as described in figure legends. We used the non-parametric, independent sample, Wilcoxon rank sum test to compare performance indices across treatment groups. Statistical testing was performed with custom code written in Matlab (Mathworks, USA). The appropriate sample size was estimated based on the standard deviation of performance indices in previous studies using the same assay (*Aso and Rubin, 2016*).

For the odor attraction and the MBON-activation assays, computing upwind displacement required us to track each fly's trajectory in time. We used the Caltech Fly Tracker (*Eyjolfsdottir et al., 2014*) to automatically extract fly trajectories from videos. Odor stimulus onset time in the arena was determined from PID measurements of odor concentration at the arena exhaust port. For the

MBON-activation assay, stimulus onset was set as the moment the LED turned on. Upwind displacement was computed as the increase in the distance from the center for each fly, relative to its location at stimulus onset, for each time-point over the entire stimulus window. The displacement for all flies in an arena experiment were then averaged before plotting and statistical testing.

## Calcium imaging

For KC data, fluorescence time-series images were first analyzed with the Suite2P analysis pipeline (**Pachitariu et al., 2016**) running in Matlab to register data and identify active single-cell regions of interest (ROIs). For MBON imaging data, ROIs were manually drawn using a custom Matlab script. For both types of experiments, average, raw fluorescence intensity for each ROI was then extracted by a separate, custom script. A background region with no labeling in each imaging field was manually defined, and background fluorescence (this consisted of the PMT offset and autofluorescence) was subtracted from all measured fluorescence values for that field. ΔF/F was computed according to the following equation ΔF/F$_i$ = (F$_i$ - F$_0$)/F$_0$.

where Fi is the fluorescence of a given cell ROI at a given time-point in a trial, and F$_0$ is the same ROI's fluorescence in an 8 s window during the baseline period on that trial, prior to odor delivery. For all plotted fluorescence traces, ΔF/F time-series data was boxcar filtered with a window-width of 0.2 s. All other analysis was done with un-filtered ΔF/F data. For making statistical comparisons, ΔF/F values during stimulus presentation were averaged over time windows as indicated in each figure.

## KC population activity decoders

The objective of this analysis was to determine whether a linear classifier can discriminate trials of a particular odor based on the KC responses. We fitted logistic regression models to predict whether or not KC activity on a given trial was evoked by a particular odor. For example, an odorA classifier received KC population activity vectors as input and then made a prediction whether the input activity was evoked by odorA/not odorA. Separate classifiers were fitted for each fly, for each odor. We used leave-one-out cross validation (LOOCV): of the 8 repeats acquired for each odor, one was left out as a test trial and the remaining trials were used to fit the model. In this way, we systematically fitted models for each combination of training and test trail sets. All plotted accuracy scores are for model predictions on test trials not used for fitting. Model weights were initialized by sampling from a distribution of weights obtained from EM connectome synapse counts (**Clements et al., 2020**). Synapse counts from a KC to an MBON were assumed to be linearly related to KC-MBON weight, and normalized to the maximum weight observed. Initial model weights were uniformly sampled from this biological distribution and then fitted without regularization.

The cost function used to estimate goodness of fit was the binary cross-entropy with a quadratic regularization, defined as

$$cost = -\frac{1}{m}\sum_{i=1}^{m} y_i \times \log\left(h_i\right) - \left(1 - y_i\right) \times \log\left(1 - h_i\right) + \frac{\lambda}{2m}\sum_{j=1}^{n} \theta_j^2$$

where $m$ is the number of training trials, $y_i$ is the correct odor label for a given trial (0 or 1), $h_i$ is the model's prediction (or probability that the input activity vector was in response to a given odor) for the same trial, $\lambda$ is the regularization constant (we used $\lambda = 1$, but this was not a sensitive parameter), $n$ is the number of neurons in a given dataset and $\theta_j$ are the weights of the neurons. For logistic regression models fitted without regularization (shown in **Figure 5B**), $\lambda$ was set to 0.

The model's prediction, $h$ was computed according to the equation

$$h = \frac{1}{1 + e^{-(X \times \theta)}}$$

here X is the $m \times n$ activity matrix for $m$ training trials and $\theta$ is the $n \times 1$ vector of weights.

## Code availability statement

The custom Matlab code used for analysis in this manuscript is publicly available at https://github.com/mehrabmodi1/Drosophila_flexible_recall (**Modi, 2023a**; copy archived at **Modi, 2023b**).

## Acknowledgements

This work was supported by the Howard Hughes Medical Institute and the National Institutes of Health (2R01DC010403-06). We thank Florin Albeanu and the Albeanu group for hosting MM for part of the time that this work was carried out, and also for engaging in helpful discussions and feedback. Robert Eifert at Cold Spring Harbor Laboratory and Steven Sawtelle, Igor Negrashov, Vasily Goncharov and others at jET, Janelia Research Campus provided vital technical support. Todd Laverty and others in the Janelia *Drosophila* resources team and Karen Hibbard from Janelia provided vital support with fly lines and media. Gudrun Ihrke and others in Project Technical Resources provided support with expression characterization. We also thank all members of the Turner and Aso groups and Eyal Gruntman, Vivek Jayaraman, Ann Hermundstad, Florin Albeanu and Priyanka Gupta for support, discussions and feedback.

## Additional information

### Funding

| Funder | Grant reference number | Author |
|---|---|---|
| Howard Hughes Medical Institute | | Yoshinori Aso<br>Glenn C Turner |
| National Institutes of Health | 2R01DC010403-06 | Glenn C Turner |

The funders had no role in study design, data collection and interpretation, or the decision to submit the work for publication.

### Author contributions

Mehrab N Modi, Conceptualization, Resources, Data curation, Software, Formal analysis, Investigation, Visualization, Writing - original draft, Writing – review and editing; Adithya E Rajagopalan, Investigation, Writing – review and editing; Hervé Rouault, Conceptualization, Formal analysis, Supervision, Investigation, Writing – review and editing; Yoshinori Aso, Conceptualization, Supervision, Funding acquisition, Investigation, Writing - original draft, Project administration, Writing – review and editing; Glenn C Turner, Conceptualization, Resources, Supervision, Funding acquisition, Writing - original draft, Writing – review and editing

### Author ORCIDs

Yoshinori Aso http://orcid.org/0000-0002-2939-1688
Glenn C Turner http://orcid.org/0000-0002-5341-2784

### Decision letter and Author response

Decision letter https://doi.org/10.7554/eLife.80923.sa1
Author response https://doi.org/10.7554/eLife.80923.sa2

## Additional files

### Supplementary files
• MDAR checklist

### Data availability
All data have been uploaded to Dryad.

The following dataset was generated:

| Author(s) | Year | Dataset title | Dataset URL | Database and Identifier |
|---|---|---|---|---|
| Turner GC | 2023 | Flexible specificity of memory in *Drosophila* depends on a comparison between choices | https://doi.org/10.5061/dryad.8931zcrtc | Dryad Digital Repository, 10.5061/dryad.8931zcrtc |

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
