## [Editor Report]

Memory recall is more precise when discrimination is required. This work in *Drosophila* shows that two related odors trigger near identical Kenyon cell responses when tested in isolation, but trigger different responses to the second odor if these are experienced in sequence within a small temporal window. The authors argue that this template comparison requires some activity downstream of Kenyon cells, that is recruited by MBONs. Overall, the experiments, building on a clever method to build "miminal memories" via optogenetically restricting the formation of memory traces in selective output compartments of the Kenyon cell (KC) axon terminals, provide very nice physiological evidence for a neural mechanism that underlies a contextual basis for the precision of memory recall.

---

## [Decision Letter]

**Decision letter after peer review:**

Thank you for submitting your article "One engram two readouts: stimulus dynamics switch a learned behavior in *Drosophila*" for consideration by *eLife*. Your article has been reviewed by 3 peer reviewers, including Mani Ramaswami as the Reviewing Editor and Reviewer #1, and the evaluation has been overseen by K VijayRaghavan as the Senior Editor.

Essential revisions:

1) There appear to be a variety of ambiguities not acknowledged in the text as well as prior work not considered in necessary depth. These are indicated or reflected in various reviewer comments. Please address all of them and provide point-by-point responses indicating how these concerns have been addressed in the revised manuscript.

2) One concern, is the claim and focus around "one engram and two responses" is both incompletely justified (e.g. see reviewer 2 comment that the boutons represent the engram and not the cells) and not intellectually useful in terms of communicating the findings to a broader audience. Figure 4, which contains the most impressive and impactful observations in the paper indicates that after the specific optogenetic learning protocol is deployed, two odors are perceived similarly (or as a common category or class of odors) in one condition but cab discriminated as two different odorants in another condition. First, at least one reviewer feels that the excitement of this figure is in the clear demonstration of. neural correlates of this contextual discrimination phenomenon – a point that is inadequately made here and somehow occluded by a pitch around "one-engram two outcomes". Second, a more solid basis for a general discussion here may be around the concept of categories. There appears to be a contextual ability to go beyond clubbing two odorants into a common category vs splitting them into distinct objects. We appreciate that the authors can decide on how they wish to discuss their data, but do ask that these suggestions/ concerns be considered as final revisions are made.

3. Please provide genetic controls requested in Reviewer 2 point d.

4. Consider if it would be useful to respond materially to the other comments.

*Reviewer #1 (Recommendations for the authors):*

1. Title of first section "Learning can support…". Implies learning is required for and/or contributes to discrimination, which would not be correct. What this really shows is that discrimination is not hindered by what appears to be "generalized" learning. A more appropriate title is needed.

2. Same page – "flies learn both generalization and discrimination tasks," What is the evidence that discrimination is learned? Perhaps it is always possible as supported by the Results section 2, showing that KC inputs in "naïve" flies have enough information for discrimination. (Though not shown, presumably this is true with MBONs too in naïve flies). It seems that the ability is not exercised unless needed. Notably (Results section 3) – learning is not associated with any increase in physiological correlates of discrimination.

3. The differences between discrimination after DAN stimulation in the two compartments could be Kenyon cells activated by one of the odorants may make more or less synapses in compartments innervated by one MBON compared to the others.

4. Figure 4 – the key: a very nice observation around which the paper's significance should really be explained/ focussed.

5. Difference between MBON a-3 and MBON g2A'1 could be due to the multiple lobes and compartments involved. The alternative, that this is an MBON dedicated to such discrimination may be true but isn't convincingly shown. In any event, a discussion of alternative possibilities is required,

6. Is the term "uncorrelated" sometimes more appropriate to use than "decorrelated?"

*Reviewer #2 (Recommendations for the authors):*

a. I am confused by the use of 'one engram' and 'single set of synapses' – 'single set' is entirely arbitrary, and what is one engram in this context? Single odor engram triggering different behavior is not novel. Several prior works have shown that a single odorant can trigger different actions in learned flies depending on either stimulus history or the fly's internal state.

b. Authors have imaged calcium dynamics in the KC cell bodies to compare if two engrams should be perceptually similar. However, Bilz et al. 2019 have shown that the KC boutons, not the cell bodies, are the units of olfactory memory engrams. I understand that imaging KC boutons will be challenging to combine with optogenetic stimulations, but this nonetheless creates a significant caveat in their approach and needs to be discussed in the manuscript.

c. Given that odorant similarity is concentration-dependent (e.g., Hallem and Carlson 2006), I am unsure how the authors chose the odor concentrations. Behavior experiments in Campbell et al. 2013 were performed in a vastly different apparatus; authors need to check if flies indeed find it difficult to discriminate between BA and PA in the current setup under standard aversive learning protocol. It would also be reassuring to see additional odor pairs to ensure the generality of their results.

d. Authors have not included any genetic control (e.g., empty split) and stimulation control (-retinal) in the behavioral experiments (Figures 1 and 6) to show the baseline response. Discrimination data in Figure 1g looks bimodal, and I wonder if discr. vs. gen. will become significant with a larger sample size. What is the rationale for the chosen sample size for these experiments? DA rescue data in Figure 1i is also incomplete as it does not contain any easy discrimination, and generalization data for γ2α'1 and α3 rescue data is entirely missing. Figure 6 includes only γ2α'1 data but no α3 activation data; α3 is an essential control for testing their predictions.

e. In Figure 4 and supplementary Figures 3 and 4, dissimilar odorants are presented as isolated pulses, not as a transition. A fairer comparison would present them as a transition to match the behavioral experience. Why were optogenetics not used for the α3 imaging experiments in Figure 3?

f. Does the starting KC weight matrix in Figure 5 match the EM connectivity data? How does the transition data from this figure look in the PC space, and how well does the decoder from Figure 2 performs on the transition data? Similarly, how well the regression model performs if it's trained purely on single pulse data?

g. Multiple papers have shown how odor transitions affect learned behavior in flies, e.g., DasGupta et al. 2014, Groschner et al. 2018, Vrontou et al. 2021. Yet, the authors have not discussed how their results support/contradict these prior models. It's an integral part of the manuscript and could point towards significant differences in experimental approach and/or intensity vs. identity coding. For example, Groschner et al. 2018 and Vrontou et al. 2021 had shown that the main difference among the KC responses during intensity discrimination is in the spike latency, not in the KC identity-I wonder if the apparent lack of discriminability in Figure 5 comes from the slow GCamp responses in the KC cell bodies.

*Reviewer #3 (Recommendations for the authors):*

The story is novel and convincing, but a little patchy in parts.

I am not convinced that Figure 4d-f should be done with shock and not DAN stimulation. The authors should attempt the equivalent optogenetic experiment for γ and α DANs.

Other less important, but noticeable points of asymmetry exist throughout the manuscript. Experiments were done with α but not γ DANs, including Figure 3 and others.

A loss of function experiment with DAN inhibition in natural learning would be appropriate and very helpful.

The dopamine biosynthesis KO experiment of Figure 1i should really be done for both populations of DANs, and ideally for easy discrimination as well as hard discrimination.

More discussion of potential mechanisms would also help.

---

## [Author Response]

Essential revisions:Reviewer #1 (Recommendations for the authors):1. Title of first section "Learning can support…". Implies learning is required for and/or contributes to discrimination, which would not be correct. What this really shows is that discrimination is not hindered by what appears to be "generalized" learning. A more appropriate title is needed.

We have now updated the section’s title to: “Precision of memory recall depends on MB compartment”. This revision also brings the section more in-line with the general comments that the manuscript should focus on how memory recall can be flexible depending on context. We have also made scattered revisions to avoid phrasing that suggests flies ‘learn to discriminate’ which gives the unintended impression that learning itself somehow makes odor cues more perceptually separate.

2. Same page – "flies learn both generalization and discrimination tasks," What is the evidence that discrimination is learned? Perhaps it is always possible as supported by the Results section 2, showing that KC inputs in "naïve" flies have enough information for discrimination. (Though not shown, presumably this is true with MBONs too in naïve flies). It seems that the ability is not exercised unless needed. Notably (Results section 3) – learning is not associated with any increase in physiological correlates of discrimination.

As mentioned above, we have revised extensively to clarify this point. In particular we revised the Introduction Lines 59-70 to clarify our meaning: flies learn a particular association and then depending on the choices available, the memory guides behavior differently, with high odor-specificity when that’s advantageous and more general when that’s best.

We do feel it is necessary (and accurate) to describe the tasks themselves as discrimination or generalization tasks. Because we often talk about how well trained flies perform these tasks, that sometimes lead to sentence constructions like ‘flies learn the hard discrimination well’ etc. But we did not intend to say that learning directly influences the sensory perception of those odors. Indeed our previous work (Figure 2 in Hige…Turner Neuron 2015) failed to find any changes in KC response patterns with optogenetic training targeted to the MB compartments, suggesting that ‘sensory perception’ of the odors does not change in these conditions. We believe our changes to the Introduction and elsewhere clear this up, but we can certainly revise further if other passages read problematically.

3. The differences between discrimination after DAN stimulation in the two compartments could be Kenyon cells activated by one of the odorants may make more or less synapses in compartments innervated by one MBON compared to the others.

In response to this and other comments, we have added a section to the Discussion Lines 465-479 that compares compartments γ2α’1 and α3 and speculates about why they differ in our observations. In this section, we have included an analysis from EM connectomics data about the number of synapses from KCs to MBONs in each compartment.

4. Figure 4 – the key: a very nice observation around which the paper's significance should really be explained/ focussed.

We thank the reviewer for their appreciation and encouragement. The pitch of the paper has been re-focussed as suggested.

5. Difference between MBON a-3 and MBON g2A'1 could be due to the multiple lobes and compartments involved. The alternative, that this is an MBON dedicated to such discrimination may be true but isn't convincingly shown. In any event, a discussion of alternative possibilities is required,

As in our response to point 3, we have added a section to the Discussion about the differences between the two compartments and how these might affect hard discrimination in them. Additionally we have re-written the section on potential mechanisms at the end of the Discussion (Lines 514-537) to discuss different possibilities.

6. Is the term "uncorrelated" sometimes more appropriate to use than "decorrelated?"

Agreed, we have replaced the word decorrelated with uncorrelated.

Reviewer #2 (Recommendations for the authors):a. I am confused by the use of 'one engram' and 'single set of synapses' – 'single set' is entirely arbitrary, and what is one engram in this context? Single odor engram triggering different behavior is not novel. Several prior works have shown that a single odorant can trigger different actions in learned flies depending on either stimulus history or the fly's internal state.

We appreciate that the manuscript needs to be re-focussed on the core observation of how similar odors are categorized. As suggested by multiple reviewers, we have updated the manuscript to not refer to engrams, extensively revising the Title, Introduction and Discussion as well as some text in the Results.

b. Authors have imaged calcium dynamics in the KC cell bodies to compare if two engrams should be perceptually similar. However, Bilz et al. 2019 have shown that the KC boutons, not the cell bodies, are the units of olfactory memory engrams. I understand that imaging KC boutons will be challenging to combine with optogenetic stimulations, but this nonetheless creates a significant caveat in their approach and needs to be discussed in the manuscript.

We revised the Discussion to raise this point as a caveat in our approach (Lines 475-479), where we address the issue that somatic signals in KCs may not reflect activity at the synapses themselves, as well as other caveats with our imaging-based approach. Since the specificity of a memory is conveyed in the pattern of activity across *many* KCs, it would be necessary to image boutons from populations of KCs and somehow connect boutons to individual cells, which is not a feasible experiment. Of course, we entirely agree that this caveat is an important point to discuss.

c. Given that odorant similarity is concentration-dependent (e.g., Hallem and Carlson 2006), I am unsure how the authors chose the odor concentrations. Behavior experiments in Campbell et al. 2013 were performed in a vastly different apparatus; authors need to check if flies indeed find it difficult to discriminate between BA and PA in the current setup under standard aversive learning protocol. It would also be reassuring to see additional odor pairs to ensure the generality of their results.

The reviewer has brought up an interesting point about differences in behavior apparatus. Although we cite Campbell 2013 for the similarity of BA and PA, the odor concentrations were chosen based on Hige 2015 where we carried out *circular arena* experiments to examine generalization between these two odors (Figure 7 in that paper). These experiments provided direct evidence that PA and BA at these concentrations are perceived similarly. We then carefully tuned odor concentrations delivered on the microscope to match the mean odor flux in the behavioral arena (see Methods).

We also note that our experimental observations provide further evidence that PA-BA discrimination is difficult at these concentrations. First, after pairing in the α3 compartment, flies perform poorly on the PA-BA behavioral discrimination, compared to a PA-EL or BA-EL discrimination. Second, even after pairing in the γ2α’1 compartment, we see depressed MBON responses to single pulses of both PA and BA, again indicative of the similarity between these two odors at the level of physiology.

d. Authors have not included any genetic control (e.g., empty split) and stimulation control (-retinal) in the behavioral experiments (Figures 1 and 6) to show the baseline response. Discrimination data in Figure 1g looks bimodal, and I wonder if discr. vs. gen. will become significant with a larger sample size. What is the rationale for the chosen sample size for these experiments? DA rescue data in Figure 1i is also incomplete as it does not contain any easy discrimination, and generalization data for γ2α'1 and α3 rescue data is entirely missing. Figure 6 includes only γ2α'1 data but no α3 activation data; α3 is an essential control for testing their predictions.

Taking these point-by-point:

Authors have not included any genetic control (e.g., empty split) and stimulation control (-retinal) in the behavioral experiments (Figures 1 and 6) to show the baseline response.

Genetic controls (empty split driver) for experiments in Figure 1 are now added as Figure 1 Supp.1. We did not carry out -retinal experiments because research on rhodopsin and *Drosophila* vision has shown that flies reared on the standard cornmeal media are able to produce all-trans retinal from the β-Carotene contained within the cornmeal (Simpson and Looger Genetics 2018). Indeed some colleagues at Janelia have observed optogenetic-induced behaviors on non-retinal added food. The empty split control maintains the comparable transgenic background with only the absence of the promoter regions and we feel is a more interpretable control.

Discrimination data in Figure 1g looks bimodal, and I wonder if discr. vs. gen. will become significant with a larger sample size. What is the rationale for the chosen sample size for these experiments?

Based on this comment, we re-examined the data used to generate these plots and have now re-computed the performance scores. The experiments plotted in Figure 1G,H are unconventional in their use of a US separated in time from odor rather than a reciprocal control. This was necessary since the dissimilar odor B is not suitable to generalize to A or A’. Hence, we had to compute a generalization score, measuring fly position during US delivery, prior to any odor delivery. We noticed that in some experiments, biased initial positions in the arena introduced a large positive or negative offset to the score. We have re-computed generalization scores after subtracting away this offset. We then assessed bi-modality of this updated dataset using Sarle’s bi-modality coefficient, which ranges from 0 to 1, with a uniform distribution showing a coefficient of 0.599. This analysis ruled out bi-modality (Sarle’s coefficients for α3 data in Figure 1G were 0.35 for discrimination and 0.43 for generalization; values for γ2α’1 data in Figure 1H were 0.35 for discrimination and 0.36 for generalization).

We chose sample size based on the standard in the field of n=12 (Aso Rubin *eLife* 2014b). This sample size was based on the n required to detect an effect size of 50% as significant with a two-sample t-test.

DA rescue data in Figure 1i is also incomplete as it does not contain any easy discrimination, and generalization data for γ2α'1 and α3 rescue data is entirely missing.

We added new dopamine (DA) rescue experiments for easy discrimination and generalization for γ2α'1 and have added this in Figure 1 Supp. 1.

DA rescue experiments for α3 could not be included due to technical difficulties. In the DA knockout background, flies locomote much less (Cichewicz…Hirsch Genes, Brain Behav. 2016), making it impossible to collect flies from the arena after the experiment to then test recall 24 hours later – we could not get them to climb out of the hole we use to extract them from the arena. γ2α'1 experiments were possible because the behavioral test is carried out immediately after training in the arena.

Figure 6 includes only γ2α'1 data but no α3 activation data; α3 is an essential control for testing their predictions.

We performed these experiments and added the results to the manuscript in Figure 6 Supp. 2. The upwind locomotion elicited by odor transitions in γ2α'1 is absent in α3, as predicted by our model.

e. In Figure 4 and supplementary Figures 3 and 4, dissimilar odorants are presented as isolated pulses, not as a transition. A fairer comparison would present them as a transition to match the behavioral experience. Why were optogenetics not used for the α3 imaging experiments in Figure 3?

To do the experiment with optogenetic reinforcement in α3, we would need a LexA driver line that labels PPL1-α3 DANs specifically and with high expression. We tried to construct split-LexA lines to specifically drive expression in these neurons, but they had very low expression levels. Hence the experiment cannot be done due to the unavailability of driver lines. We have added this explanation (Lines 331-334) and example images of split-LexA lines we tried (Figure 4 Supp. 4)

f. Does the starting KC weight matrix in Figure 5 match the EM connectivity data? How does the transition data from this figure look in the PC space, and how well does the decoder from Figure 2 performs on the transition data? Similarly, how well the regression model performs if it's trained purely on single pulse data?

In the original Figure 5B,C we arbitrarily assigned the weights; we have now replaced this using KC weights sampled from the distribution obtained from connectomics, thank you for the suggestion. This analysis supports the original conclusion that KC responses to single pulses do not predict MBON responses to odor transitions.

We have added a Figure 5 Sup Figure 1 to address the points here. The PCA plots of transition data are now shown in Figure 5 Supp. 1A. This visualization shows that responses to the second odor in the transition are quite similar to responses to isolated pulses. Indeed decoders trained on single pulses (as in Figure 2) distinguish odors in a transition almost as accurately as the isolated pulses Figure 5 Supp. 1B,C. This is consistent with Figure 5B in the main text which shows that a decoder trained on single pulses fails to distinguish between A-A’ and A’-A transitions. Together these results support the conclusion that KC activity patterns do not reflect the odor transition responses we observe in MBON.

We have also revised the Results section to clarify the different decoder analyses, and added schematics to Figure 5B,C to more clearly indicate the different training procedures.

g. Multiple papers have shown how odor transitions affect learned behavior in flies, e.g., DasGupta et al. 2014, Groschner et al. 2018, Vrontou et al. 2021. Yet, the authors have not discussed how their results support/contradict these prior models. It's an integral part of the manuscript and could point towards significant differences in experimental approach and/or intensity vs. identity coding. For example, Groschner et al. 2018 and Vrontou et al. 2021 had shown that the main difference among the KC responses during intensity discrimination is in the spike latency, not in the KC identity-I wonder if the apparent lack of discriminability in Figure 5 comes from the slow GCamp responses in the KC cell bodies.

Thank you for pointing out some gaps in our discussion of the literature. Groschner 2018 was referred to in the Discussion in the original manuscript, but we now also discuss DasGupta 2014 and Vrontou 2021. These three studies show that KCs act as leaky integrators of odor-evoked excitation, showing different spike latencies for different stimulus conditions. It is possible that very similar odors could activate the same KCs, but with different latencies, that might not be resolved by calcium imaging. We have included this point in the revised Discussion (lines 475-479), where we write that different KC subtypes could have different integrative properties that might contribute to discriminating, and cite these studies.

We do want to emphasize here that although KC response patterns are extensively overlapping for PA and BA, they do in fact contain enough information to discriminate between single pulses of these odors even with the low temporal resolution of calcium imaging. The issue is rather that the plasticity rule in this compartment is such that the extensive overlap between KC patterns results in depressed responses to both odors when we follow the downstream MBON.

Reviewer #3 (Recommendations for the authors):The story is novel and convincing, but a little patchy in parts.

We have extensively re-written the paper, focussing on odor-categorization rather than on changing recall of a single engram. We hope this improves the manuscript’s flow.

I am not convinced that Figure 4d-f should be done with shock and not DAN stimulation. The authors should attempt the equivalent optogenetic experiment for γ and α DANs.

See our reply to Reviewer #2, point e for a detailed response, and explanation in the main text (Lines 331-334). Briefly, this experiment cannot be done using optogenetics due to the unavailability of driver lines, despite our efforts to produce them (see Figure 4 Supp. 4).

Other less important, but noticeable points of asymmetry exist throughout the manuscript. Experiments were done with α but not γ DANs, including Figure 3 and others.

We think the reviewer meant to say experiments in Figure 3 were not done for the α3 compartment. In Figure 3 our goal was to find how neural activity supports distinct learned responses for the two similar odors in the hard discrimination task. In Figure 1 we had shown that this was a property of the γ2α’1 compartment, but in Figure 3 our imaging showed that, rather than discriminating, MBON-γ2α’1 generalizes across odors A and A’. Since training targeted to α3 does not support significant performance on the hard discrimination task, we did not characterize MBON-α3 with single odor pulses. However, we note that MBON-α3 responses to odor transitions (Figure 4) clearly show similar levels of depression for both A and A’ irrespective of the order of the transition.

We note that for other points of α3/γ2α’1 asymmetry – particularly Figure 1 and Figure 6 – we have now carried out the missing experiments.

A loss of function experiment with DAN inhibition in natural learning would be appropriate and very helpful.

We considered this experiment, but it is likely that interpretation will be very difficult. There are 9 compartments involved in aversive learning, so if any other compartment has the ability to support fine discrimination, the effects of loss of function experiment in γ2α’1 would be masked. Indeed there are very few loss of function experiments targeting individual compartments in the literature in general, likely because of the redundancy between some of the 15 compartments.

The dopamine biosynthesis KO experiment of Figure 1i should really be done for both populations of DANs, and ideally for easy discrimination as well as hard discrimination.

We thank the reviewer for this suggestion, also raised above (Ref 2 point d). We have now done the suite of easy/hard/generalization experiments in flies with dopamine biosynthesis confined to DAN PPL1-γ2α’1 and present the results in Figure 1 Supp. 1. However, we have not been able to do these experiments with the α3 compartment due to technical limitations. Flies lacking DA throughout the rest of the nervous system locomote much less, making it hard to train and collect flies and then test recall 24 hours later in order to do these experiments targeting α3. It was possible γ2α'1 only because the behavioral test is carried out immediately after training in the arena.

More discussion of potential mechanisms would also help.

We have added a clearer description of the possible mechanisms to the revised Discussion (Lines 514-53).